# Integrating Reanalysis and Satellite Cloud Information to Estimate Surface Downward Long-Wave Radiation

**Francis M. Lopes** [1,*] , **Emanuel Dutra** [1,2] **and Isabel F. Trigo** [1,2]

1 Instituto Dom Luiz (IDL), Faculty of Sciences, University of Lisbon, Campo Grande, 1749-016 Lisbon, Portugal; emanuel.dutra@ipma.pt (E.D.); isabel.trigo@ipma.pt (I.F.T.)
2 Instituto Português do Mar e da Atmosfera (IPMA), Rua C do Aeroporto, 1749-077 Lisbon, Portugal
* Correspondence: fmtlopes@fc.ul.pt

**Abstract:** The estimation of downward long-wave radiation (DLR) at the surface is very important for the understanding of the Earth's radiative budget with implications in surface–atmosphere exchanges, climate variability, and global warming. Theoretical radiative transfer and observationally based studies identify the crucial role of clouds in modulating the temporal and spatial variability of DLR. In this study, a new machine learning algorithm that uses multivariate adaptive regression splines (MARS) and the combination of near-surface meteorological data with satellite cloud information is proposed. The new algorithm is compared with the current operational formulation used by the European Organization for the Exploitation of Meteorological Satellites (EUMETSAT) Satellite Application Facility on Land Surface Analysis (LSA-SAF). Both algorithms use near-surface temperature and dewpoint temperature along with total column water vapor from the latest European Centre for Medium-range Weather Forecasts (ECMWF) reanalysis ERA5 and satellite cloud information from the Meteosat Second Generation. The algorithms are trained and validated using both ECMWF-ERA5 and DLR acquired from 23 ground stations as part of the Baseline Surface Radiation Network (BSRN) and the Atmospheric Radiation Measurement (ARM) user facility. Results show that the MARS algorithm generally improves DLR estimation in comparison with other model estimates, particularly when trained with observations. When considering all the validation data, root mean square errors (RMSEs) of 18.76, 23.55, and 22.08 W·m$^{-2}$ are obtained for MARS, operational LSA-SAF, and ERA5, respectively. The added value of using the satellite cloud information is accessed by comparing with estimates driven by ERA5 total cloud cover, showing an increase of 17% of the RMSE. The consistency of MARS estimate is also tested against an independent dataset of 52 ground stations (from FLUXNET2015), further supporting the good performance of the proposed model.

**Keywords:** downward surface long-wave radiation; machine learning; multivariate adaptive regression splines; EUMETSAT LSA-SAF; ECMWF-ERA5

## 1. Introduction

The downward long-wave radiation (DLR hereafter), defined as the irradiance reaching the surface in the infrared range between 4 and 100 μm, is an essential component of the Earth's surface radiation budget [1–3]. DLR has a high dependency with the vertical profiles of atmospheric temperature, water vapor (the largest contributor to the greenhouse effect [4]), and cloud cover. Therefore, accurate estimations of DLR are important for a wide range of applications dealing with climate variability [5]. Since DLR is a key component of the land surface radiative balance, it is essential to model and estimate the land surface turbulent fluxes (latent and sensible), which are relevant to predict the effects that climate and land use changes have on water resources, ecosystems, and the agricultural sector [6]. Naud and Miller [7] have reported DLR high sensitivity to changes in water vapor in high-elevation regions, which are among the most sensitive regions to future climate change. This is of particular relevance, since in such remote regions there is usually

a lack of measured DLR and, therefore, an absence of information for determination of possible warming rates triggers. There are other applications in which DLR estimates are essential, such as for the design of passive cooling systems in buildings, where measured values of DLR are usually absent [8].

In the past decades, several research works have been performed to estimate DLR recurring to empirical formulations. The earlier studies were conducted only for clear-sky conditions (e.g., [9–13]). Most of these formulations considered the two main modulators of DLR in clear-sky conditions: temperature and moisture. Recently, several studies explored the effects of clouds on the sky apparent emissivity and therefore on DLR, introducing cloud fraction parameterizations to estimate DLR under all-sky conditions (e.g., [14–18]). These new models brought more flexible and complex approaches to estimate DLR under different sky conditions, considering atmospheric profile databases, semi-empirical or multiple regression methods (e.g., [2,14,16,19,20]), hybrid systems that combine physical models and remotely sensed data (e.g., [17]), and, more recently, machine-learning techniques (e.g., [21–30]). The latter provided the capability to handle complex nonlinear statistical problems, particularly the nonlinear relation between DLR and its main modulators. These studies have shown satisfactory results when combining remote sensed information with machine-learning algorithms for DLR estimation, such as extremely randomized trees (ERT) [24], random forest (RF) [25,27,29], and artificial neuron network (ANN) [30], surpassing the previous simpler methods. In particular, to the present date, only a restricted number of machine-learning studies have applied MARS for the estimation of DLR. For instance, Feng et al. [21] demonstrated the MARS potential for the determination of daily and monthly DLR values under all-sky conditions, including regions of high and low altitude. However, despite their good results, the authors underlined the need for reducing the obtained bias and model overfitting. Zhou et al. [23,28] included MARS as part of a hybrid system that performed estimates of DLR under clear-sky conditions using the moderate-resolution imaging spectroradiometer (MODIS) thermal infrared bands top of the atmosphere radiances and surface measurements of DLR. Although the proposed methodology (entirely dependable on satellite and ground data without the use of NWP models) led to some deviations in the results, these showed an overall good performance of the method when remote sensing-based DLR estimations are used. In another study, Jung et al. [31] used several machine-learning methods, including MARS, to combine energy flux measurements acquired from FLUXNET eddy covariance towers with MODIS and meteorological data and to produce the FLUXCOM dataset. The resulting FLUXCOM estimates were found to be suitable for the quantification of global land–atmosphere interactions and land surface model simulations benchmarking. Although there is no definitive regression model technique to be used for all situations, MARS algorithms have proven to have a good bias-variance trade-off (with fairly low bias and variances), being flexible enough for modelling non-linearity and handling a large number of input variables (i.e., more than two variables), while in other simpler models such dimensionality generates problems [32].

In the present work, we present a novel and synergetic approach that uses MARS to combine the European Centre for Medium-range Weather Forecasts (ECMWF) reanalysis with ground, and remote sensed information to estimate DLR hourly values under all-sky conditions. Although it is not possible to directly infer DLR from remotely sensed observations under overcast conditions [10], the combination of satellite information and numerical weather models has the advantage of providing accurate DLR values over large areas [14]. Moreover, this combination of data sources allows the estimate of DLR in remote locations of difficult man-made access in which the installation of measuring equipment is not viable. Despite improvements, the precise determination of DLR following these approaches has some degree of dependency with factors that hinder their accuracy. For instance: (i) pure empirical methods are limited due to particular calibration conditions (frequently applicable to clear-sky conditions only); (ii) physical models are dependent on the quality and availability of the atmospheric profile databases used; and (iii) satellite derived-data often

lack accuracy, especially because top-of-atmosphere observations are only indirectly related with DLR. The latter generally provides information on cloud fraction or type, which clearly influences DLR. However, more specific variables affecting DLR estimations, particularly cloud-base properties (height, temperature, and emissivity), are difficult to measure or to model (e.g., [33–35]). It is therefore fundamental to create a robust synergistic approach that combines the use of ground and remote sensing observations with numerical weather prediction (NWP) models to quantify accurately DLR at the surface.

The starting point of this study is the semi-empirical model presented by Trigo et al. [14], named here as the LSA-SAF model, which is currently operational in near real-time by the European Organization for the Exploitation of Meteorological Satellites (EUMETSAT) Satellite Application Facility on Land Surface Analysis (LSA-SAF). More details regarding LSA-SAF, and respective products, are available online (https://www.eumetsat.int/lsa-saf; accessed on 31 March 2022). The LSA-SAF currently provides DLR estimates as 30-min instantaneous fluxes (product MSDSLF, LSA-204) and as daily averages (product DIDSLF, LSA-206), both covering the Meteosat Second Generation (MSG) disk. The LSA-SAF model is based on a small set of atmospheric variables, namely total column water vapor, near-surface temperature, and near surface dewpoint temperature, being calibrated separately for clear and cloudy-sky conditions. It uses cloud cover information, provided by the spinning enhanced visible and infrared imager (SEVIRI) on board of the MSG satellite series, to establish the sky classification. The MSG main optical payload SEVIRI was built for the purpose of providing several NWP and climate applications over the European and African regions [36], although South America is also covered. The LSA-SAF model calibration was based on downward infrared flux simulations provided by the Moderate Resolution Atmospheric Transmittance and Radiance Code (MODTRAN-4, [37]), using the Thermodynamic Initial Guess Retrieval (TIGR) atmospheric-profile database presented by Chevallier et al. [38]. The latter includes a subset of temperature and humidity profiles from the ECMWF ERA-40 reanalysis covering a wide range of different atmospheric states (classified into dry cold, dry warm and moist) fed to the MODTRAN-4 to calculate DLR at the surface. The LSA-SAF algorithm was derived by adjusting such DLR estimates to a semi-empirical function using uniquely total water vapor content and screen variables (near surface temperature and dew point). The model was then validated using an independent dataset, which showed that the algorithm is able to reproduce reasonably well DLR values at the surface under clear and cloudy skies, with low bias and root mean square errors. Furthermore, Trigo et al. [14] showed that combining satellite cloud information with bulk and screen variables led to competitive results when compared with ECMWF estimates, a result that highlighted the essential role of clouds in DLR. The LSA-SAF simple model was therefore shown to be a viable option to derive DLR over large areas. Nevertheless, after a long period in operations, the systematic comparison of DLR estimates from the LSA-SAF model with station observations suggests that there is still room for improvements. More details regarding the LSA-SAF model are available in the product user manual [39].

This work aims at establishing a simple and improved DLR algorithm for operational purposes. The resulting DLR product should be compatible with MSG cloud information to guarantee consistency among the different LSA-SAF products (e.g., downward solar radiation). To this end, a new and more flexible formulation is proposed to estimate DLR, making use of a machine learning algorithm based on multivariate adaptive regression splines (MARS). This new approach combines recursive partitioning and spline fitting in the form of a series of step (or hinge) functions and knots, as demonstrated by Freidman [40], replacing simpler regression methods as the one performed in the original calibration of the LSA-SAF algorithm. Similarly to Trigo et al. [14], the proposed method treats clear and cloudy conditions separately, thus allowing the training of two different models using ground measured DLR fluxes from several in situ stations as reference. Cloud classification is based on satellite (MSG) observations, while each model uses ERA5 total water column vapor and screen variables (2-metre temperature and dew point) as independent variables. The validation of the new methodology consists of the assessment of

DLR estimates from MARS against another set of ground stations previously excluded from the training process. The in situ measurements are provided by the Baseline Surface Radiation Network (BSRN, [41]) and the Atmospheric Radiation Measurement (ARM, [42]) user facility, all within the MSG-disk (as described in Section 2.1). Additionally, to assess the consistency of the proposed methodology, MARS estimates are also validated against an independent spatiotemporal dataset of 52 ground stations from FLUXNET2015. In this analysis, besides DLR estimates from the MARS model, other model estimates are also considered for comparison purposes, including the operational LSA-SAF model, a new LSA model calibrated with ERA5 and measured data, and ERA5 radiation fluxes. Additionally, the MARS and LSA models are also driven by ERA5 cloud information to assess the added value of the MSG cloud information in the estimation of DLR. More details regarding MARS application, as well as a description of other models used in this study, are presented further on in Section 2.2.

The remainder of this paper is structured as follows: in Section 2, data and methods are presented, including a description for the LSA and MARS models; Section 3 provides the results for the validation of the MARS model against in situ measurements and other models estimates within the MSG-disk; in Section 4, a discussion for the obtained results is presented; while conclusions and future perspectives are given in Section 5; additional information related to the analysis is added at the end of this paper in Appendix A, Appendix B, Appendix C, Appendix D.

## 2. Methodology

### 2.1. Observations and Reanalysis

The BSRN [41] has been operational since 1995, as part of the world climate research program, supported by the World Meteorological Organization (WMO) and others. As a network of surface radiation monitoring, significant improvements have been achieved over the last decades with the simultaneous increase of globally scattered stations and the quality of ground measured data. Although there are currently 57 operational stations installed over different surface types, measured data from a total of 77 stations is freely available (https://bsrn.awi.de/; accessed on 31 March 2022), with quality control procedures.

In this work a total of 22 BSRN stations were used (Figure 1a), being located within the MSG-disk (i.e., longitude/latitude within $+/- 75°$E/N), and with available data within the 16-year period from 2004 to 2019. Three BSRN stations located in the MSG-disk are not considered for analysis, either because of representative issues of the measurements (Izaña, in Tenerife, Spain) or due to complete absence of data (Ilorin, Nigeria; and Rolim de Moura, Brazil). Izaña, is a relatively high-altitude station (2372.9 m), often above the clouds that cover most of the island; under these conditions, the MSG pixel is usually correctly classified as "cloud covered", but DLR observations are characteristic of "clear sky" and therefore inconsistent with the satellite information. In addition to the BSRN stations, the Niamey station (13.4773°N; 2.1758°W) from the ARM user facility was also included (60-s downwelling irradiances from the sky radiation sensor SKYRAD60S). Niamey station was selected due to its particular local atmospheric features, in particular the aerosol load [43], which can be in the form of severe dust events, such as desert storms. More details concerning Niamey mobile facility and radiation observations are available in Sengupta et al. [44].

Although quality control procedures have been previously applied, several outliers were found across the 23 stations (see Table 1), as well as a period of about 5 months in SMS station with measurements made with malfunctioning equipment. Accordingly, the corresponding sets of data were removed from the analysis, increasing the number of gaps in the in situ observations. There is a diverse amount of temporal coverage across all stations. For instance, the station that has the most complete list of records (TAM) only has 0.83% of missing data during the 16-year period, while the station with the lowest number of records (BUD) has 99.49% of missing data. For comparison purposes, in this analysis all DLR observations were temporally aggregated to hourly frequency.

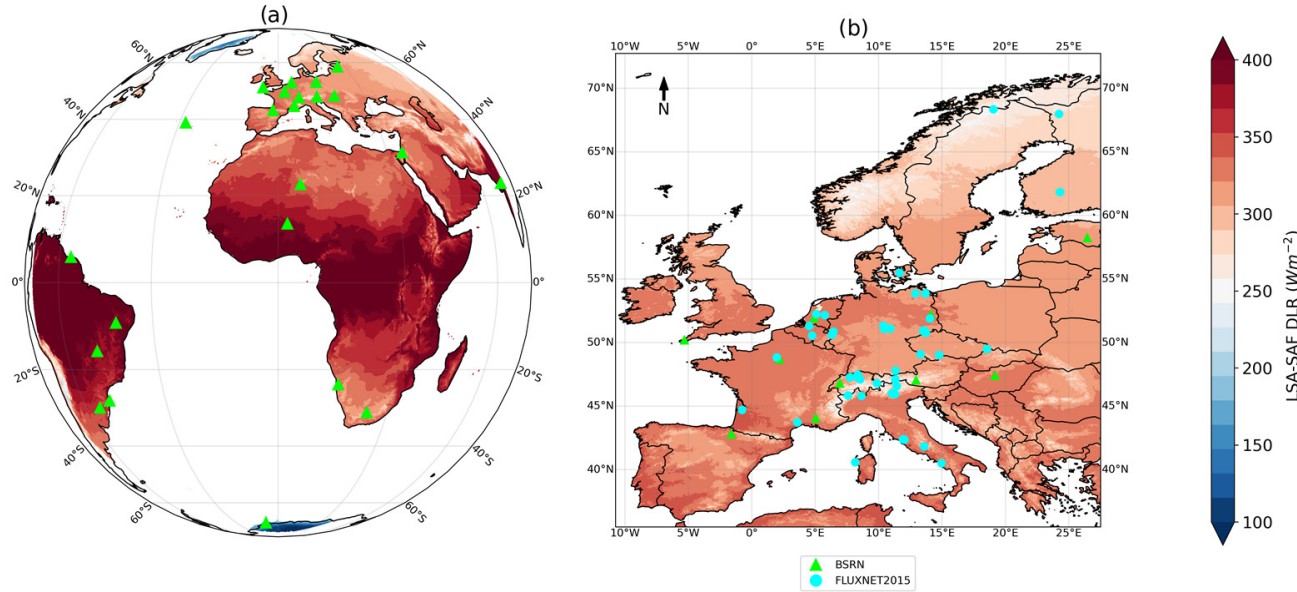

**Figure 1.** (**a**) Example of the annual mean (2020) downward long-wave radiation (DLR) at the surface estimated with the LSA-SAF operational algorithm within the Meteosat Second Generation (MSG) disk, including the location for each of the 23 Baseline Surface Radiation Network (BSRN) ground stations marked by the green triangles; (**b**) Zoom of the European region, including both BSRN and also the FLUXNET2015 48 ground stations marked by the cyan circles.

**Table 1.** List of 23 stations used within the Meteosat Second Generation (MSG) disk for validation purposes of estimated downward long-wave radiation (DLR) at the surface. The name, acronym, network of origin, location, geographical coordinates (°), elevation (m), availability (total number of years available between 2004 and 2019), and annual mean DLR (W·m$^{-2}$) for each station is shown.

| Station | Acronym | Network | Location | Latitude and Longitude (°) | Elev. (m) | Avail. (Years) | Annual DLR (W·m$^{-2}$) |
|---|---|---|---|---|---|---|---|
| Brasília | BRB | BSRN | Brazil | 15.60°S; 47.71°W | 1023 | 7.12 | 364.45 |
| Budapest | BUD | BSRN | Hungary | 47.43°N; 19.18°E | 139 | 0.08 | 373.82 |
| Cabauw | CAB | BSRN | Netherlands | 51.97°N; 4.93°E | 0 | 14.69 | 323.69 |
| Camborne | CAM | BSRN | U.K. | 50.22°N; 5.32°W | 88 | 11.64 | 324.57 |
| Carpentras | CAR | BSRN | France | 44.08°N; 5.06°E | 100 | 14.15 | 321.74 |
| Cener | CNR | BSRN | Spain | 42.82°N; 1.60°W | 471 | 10.28 | 321.71 |
| De Aar | DAA | BSRN | South Africa | 30.67°S; 23.99°E | 1287 | 6.25 | 303.88 |
| Eastern North Atlantic | ENA | BSRN | Azores | 39.09°N; 28.03°W | 15.2 | 1.00 | 359.34 |
| Florianopolis | FLO | BSRN | Brazil | 27.61°S; 48.52°W | 11 | 5.70 | 386.40 |
| Gandhinagar | GAN | BSRN | India | 23.11°N; 72.63°E | 65 | 1.58 | 401.45 |
| Gobabeb | GOB | BSRN | Namibia | 23.56°S; 15.04°E | 407 | 7.54 | 338.67 |
| Neumayer | GVN | BSRN | Antarctica | 70.65°S; 8.25°W | 42 | 14.89 | 216.87 |
| Niamey | NIM | ARM | Africa | 13.48°N; 2.18°E | 223 | 1.02 | 392.11 |
| Lindenberg | LIN | BSRN | Germany | 52.21°N; 14.12°E | 125 | 13.99 | 315.06 |
| Palaiseau | PAL | BSRN | France | 48.71°N; 2.21°E | 156 | 15.63 | 322.61 |
| Paramaribo | PAR | BSRN | Suriname | 5.81°N; 55.22°W | 4 | 0.58 | 421.16 |
| Payerne | PAY | BSRN | Switzerland | 46.82°N; 6.94°E | 491 | 15.70 | 315.05 |
| Petrolina | PTR | BSRN | Brazil | 9.07°S; 40.32°W | 387 | 7.56 | 386.86 |
| Sede Boqer | SBO | BSRN | Israel | 30.86°N; 34.78°E | 500 | 7.49 | 332.86 |
| São Martinho da Serra | SMS | BSRN | Brazil | 29.44°S; 53.82°W | 489 | 6.04 | 327.19 |
| Sonnblick | SON | BSRN | Austria | 47.05°N; 12.96°E | 3109 | 6.28 | 249.07 |
| Tamanrasset | TAM | BSRN | Algeria | 22.79°N; 5.53°E | 1385 | 15.88 | 330.70 |
| Toravere | TOR | BSRN | Estonia | 58.25°N; 26.46°E | 70 | 15.70 | 308.71 |

To further improve and reinforce the proposed validation procedure, an independent dataset of ground observations from the FLUXNET2015 network has also been considered (https://fluxnet.org/data/fluxnet2015-dataset/, [45]; accessed on 31 March 2022). To this end, half-hourly measurements from 52 ground stations were aggregated to hourly values

and then used for validation purposes. The selected measured variable was the incoming longwave radiation (LW_IN_F_MDS), which is gap-filled using the Marginal Distribution Sampling (MDS) method, i.e., taking into account observations made under similar meteorological, physical and temporal conditions [46]. It should be noted that only 52 stations (Table A8) were eligible for the present study, since these needed to follow several requirements at the same time, i.e., to be within the MSG-disk, within the 2004–2015 period, and should have representative values. Regarding the latter, during a quality control check it was observed that several stations had different measuring periods with MDS gap-filling applied, being characterized by a "poor-quality" flag (i.e., the lowest level) and, therefore, are removed from the analysis. In this context, out of 198 stations globally available, only 52 were suitable. Moreover, out of these, 48 stations are located in Europe (Figure 1b). More details regarding the FLUXNET2015 stations and respective validation results for the MARS model are shown in Appendix D.

In addition to ground measured data, several fields with hourly frequency of the most recent ECMWF reanalysis, ERA5 [47], were extracted from the Copernicus Data Store (CDS). The fields include total column water vapor (*tcwv*, mm), 2-metre temperature (*t2m*, K), 2-metre dewpoint temperature (*d2m*, K), total cloud cover (*tcc*), and downwelling surface thermal radiation (*strd* or DLR, W·m$^{-2}$), with the latter being produced through the McRad radiation scheme [48]. For comparison purposes, the ERA5 fields were interpolated to the measuring station's location following a nearest neighbor approach. It is worth noting that, for the evaluation of MARS and LSA models against observations, both *t2m* and *d2m* temperatures were adjusted to each measuring altitude considering a reference temperature lapse rate of $-6.5$ K/km [49]. Similarly, ERA5 radiation fluxes were adjusted considering a correction factor of $-2.8$ W/m$^2$ per 100 m [13].

Cloud mask information retrieved from the SEVIRI sensor on board MSG is also used in this work (at 15-min frequency) for the definition of sky conditions. As described by Derrien and Gléau [50], the MSG cloud mask was developed by the LSA on support to Nowcasting and Very Short-Range Forecasting (NWC-SAF, https://www.eumetsat.int/nwc-saf; accessed on 31 March 2022), allowing to identify cloud free areas where different products can be computed (e.g., total precipitable water, land, or sea surface temperatures), as well as cloudy areas from which other products can be derived (e.g., cloud type or cloud top temperature/height). Several research works have shown the added value of the MSG information for cloud detection (e.g., [14,51]). In particular, Trigo et al. [14] showed an overall good performance of the MSG cloud mask in cloud identification during the validation of DLR estimates. However, despite the satisfactory results, these authors also observed that, in regions under high aerosol load, the accuracy of the satellite cloud mask could contribute to a lower performance of the proposed method. In the context of DLR estimation, the present study uses cloud fraction (denoted *cf*) retrieved from the SEVIRI sensor for the training and evaluation of both MARS and LSA algorithms. For this purpose, 15-min cloud mask data is aggregated to hourly *cf* using an hourly rolling mean. The procedure allows to select pure situations of clear (*cf* = 0) and cloudy (*cf* = 1) conditions during a particular hour.

### 2.2. Models

Two algorithms that estimate DLR are evaluated in this study: (i) the current operational semi-empirical algorithm used by LSA-SAF (referred as LSA hereafter) and (ii) a new MARS algorithm, a more flexible approach for the definition of the different atmospheric states under which DLR is calculated. The resulting models (LSA and MARS) are both driven by ERA5 atmospheric conditions (*tcwv*, *t2m*, and *d2m*), satellite cloud cover from MSG/SEVIRI observations, and are calibrated using DLR observations from the BSRN and ARM stations. Additionally, estimates of DLR from ECMWF-ERA5 reanalysis (ERA5) and of the current LSA-SAF operational product (LSA_OPER) are also considered in the analysis. For completeness, to assess the value of using satellite cloud information to calculate DLR with both algorithms, LSA and MARS models were also applied using ERA5

total cloud cover (denoted LSA* and MARS*, respectively) instead of the MSG cf. Table 2 resumes the key characteristics of the models used in the analysis, including the MARS algorithm for the MARS and MARS* models, the LSA-SAF for the LSA and LSA* models, and the ERA5 reanalysis for the ERA5 model.

**Table 2.** List of models used in the analysis, including respective predictors, predictands, and cloud information, for the training and evaluation periods.

| | **Training** | | | | **Evaluation** | |
|---|---|---|---|---|---|---|
| **Model** | **Predictors** | **Cloud Info.** | **Predictand** | **Period** | **Predictors** | **Cloud Info.** |
| MARS LSA | *tcwv*, *t2m*, *d2m* (ERA5) | *cf* (MSG) | DLR (BSRN, ARM) | 2004–2019 [1] | *tcwv*, *t2m*, *d2m* (ERA5) | *cf* (MSG) |
| MARS* LSA* | | | | | *tcwv*, *t2m*, *d2m* (ERA5) | *tcc* (ERA5) |
| LSA_OPER | *tcwv*, *t2m*, *d2m* (ERA-40) | *tcc*(ERA-40) | DLR (MODTRAN-4) | 1992–1993 | *tcwv*, *t2m*, *d2m* (ECMWF operational NWP) | *cf* (MSG) |
| ERA5 | - | - | - | - | - | - |

[1] Random selection of 6 months of data from each station.

The LSA-SAF algorithm is presented in detail by Trigo et al. [14] and resumed in Appendix A. The piecewise regression approach used by LSA-SAF is based on three classes of atmospheric profiles, independent for clear and cloudy conditions, which were manually selected (Table A1). Similarly, MARS [40,52] is based on a weighted sum of piecewise functions, also known as basis functions, in which the MARS additive model follows the recursive partitioning regression form, as described by Friedman [40]. The resulting regression coefficients are then adjusted to find the best fitting to the data. The selection of the basis functions is a fundamental process in MARS: this consists of an automatic procedure following a two-stage building process, established by performing a forward and a backward step. Compared with the LSA-SAF algorithm, the automatic procedure to establish the piecewise regression in MARS is a key advantage. In this study, the MARS algorithm available in the py-earth python package (version 0.1.0, https://github.com/scikit-learn-contrib/py-earth; accessed on 31 March 2022) is used. As in the case of the LSA algorithm, two MARS sub-models are trained for clear and cloudy conditions, respectively, considering "pure types" identified by the MSG *cf* (0 or 1); the all-sky DLR is computed following Equation (A4).

Both LSA and MARS models were calibrated with the same subsets of data independently for clear and cloudy conditions, using the MSG *cf*. The models used ERA5 *tcwv*, *t2m*, and *d2m* as predictors, and observations (BSRN and ARM) of DLR as predictand. Since there are large differences in data availability for each station (Table 1), the models were calibrated with a randomly selected sample of 40% of the full-time series in each station limited to a maximum of 6 months of data. The procedure allowed us to avoid the dominance of some stations with longer periods in the training dataset. In an initial phase, the MARS model was also tested with different combinations between the three predictors (*tcwv*, *t2m*, *d2m*). The results (not shown) indicated that the use of the three predictors provides the best outputs, although *tcwv* and *t2m* alone could already generate reasonable results. Moreover, the addition of the *cf* was also tested as an explicit input (predictor) in MARS, i.e., as an alternative to the two sub-models for clear and cloudy conditions. This approach did not perform as well as for the two sub-models (not shown), most likely due to the binary nature of the cloud information, which is not optimal for the MARS model. Considering these preliminary tests, and for consistency with LSA, it was decided to keep the three predictors in MARS and two independent sub-models, i.e., one for clear and another for cloudy conditions.

The training of the LSA-SAF algorithm follows Trigo et al. [14], described briefly in Appendix A, where the resulting calibrated parameters are shown in Table A2. For MARS, a repeated k-fold cross-validation procedure was considered during the training phase. The process involves the performance of repeated cross-validation procedure several times and calculation of the mean result across all folds. For the present study, a 10-fold was considered, since this value is typically used in machine learning models (e.g., [53–55]), being found to provide a good trade-off of low computational cost and low bias. The results of the training of both MARS and LSA models for clear and cloudy conditions are presented in detail in Appendix B, showing the respective model performance in the training and validation datasets.

*2.3. Evaluation Metrics*

The performance of the different DLR estimates obtained from each model is assessed through a series of conventional error metrics, such as bias ($\mu$), the root mean square error (RMSE), standard deviation of the error or unbiased root mean square error ($\sigma$), and the temporal correlation coefficient (R):

$$\mu = \frac{1}{N} \sum_{i=1}^{N} d_i, \ d_i = y_i - o_i \tag{1}$$

$$\text{RMSE} = \frac{1}{N} \sqrt{\sum_{i=1}^{N} (d_i)^2}, \tag{2}$$

$$\sigma = \frac{1}{N-1} \sqrt{\sum_{i=1}^{N} (d_i - \mu)^2}, \tag{3}$$

$$R = \frac{\sum_{i=1}^{N} (y_i - \overline{y})(o_i - \overline{o})}{\sqrt{\sum_{i=1}^{N} (y_i - \overline{y})^2 \sum_{i=1}^{N} (o_i - \overline{o})^2}}, \tag{4}$$

where $y_i$ is the modelled value of the i-th sample (N is the number of samples), $o_i$ is the corresponding reference value, $d_i$ is the difference between the modelled and reference, with the overbar representing the temporal mean of a variable.

**3. Results**

*3.1. Model Evaluation*

The models were evaluated considering the whole DLR observational dataset between 2004 and 2019; statistics obtained for the independent datasets used in model training and verification are shown in Appendix B. Since there is a large range of the temporal coverage among the stations, the evaluation was performed following two approaches: (i) merging all station data before the evaluation, computing the overall metrics, and displaying the results as density scatter plots; and (ii) computing the evaluation metrics for each station independently, displaying the distributions of the metrics as boxplots.

The overall performance of the models, i.e., merging all stations before the evaluation, is depicted in Figure 2. The corresponding density scatter plot for each model and sky condition (i.e., clear, cloudy, and all-sky) shows the model DLR as function of the observations along with the different metrics. The absolute biases are always below 2 W·m$^{-2}$ for both LSA and MARS models. This is expected since the model's training aims at the minimization of systematic differences. For the case of ERA5, biases are also small under clear-sky, however these grow to $-14.54$ W·m$^{-2}$ in cloudy conditions, which result in an all-sky bias of $-5.25$ W·m$^{-2}$. The root mean square error is dominated by the error variability (standard deviation of the error) in all models. This can be primarily attributed to temporal/spatial variability that is not captured by ERA5 or by the ERA5 predictors used in LSA and MARS. In terms of the RMSE, MARS has the best performance in all the different sky conditions, with an error of 18.76 W·m$^{-2}$ under all-sky, being followed by LSA with 20.24, ERA5 with 22.08, and LSA_OPER with 23.55 W·m$^{-2}$. Moreover, the linear

correlations are always above 0.91 in all models and sky conditions, with MARS showing a consistently better performance, although differences are small.

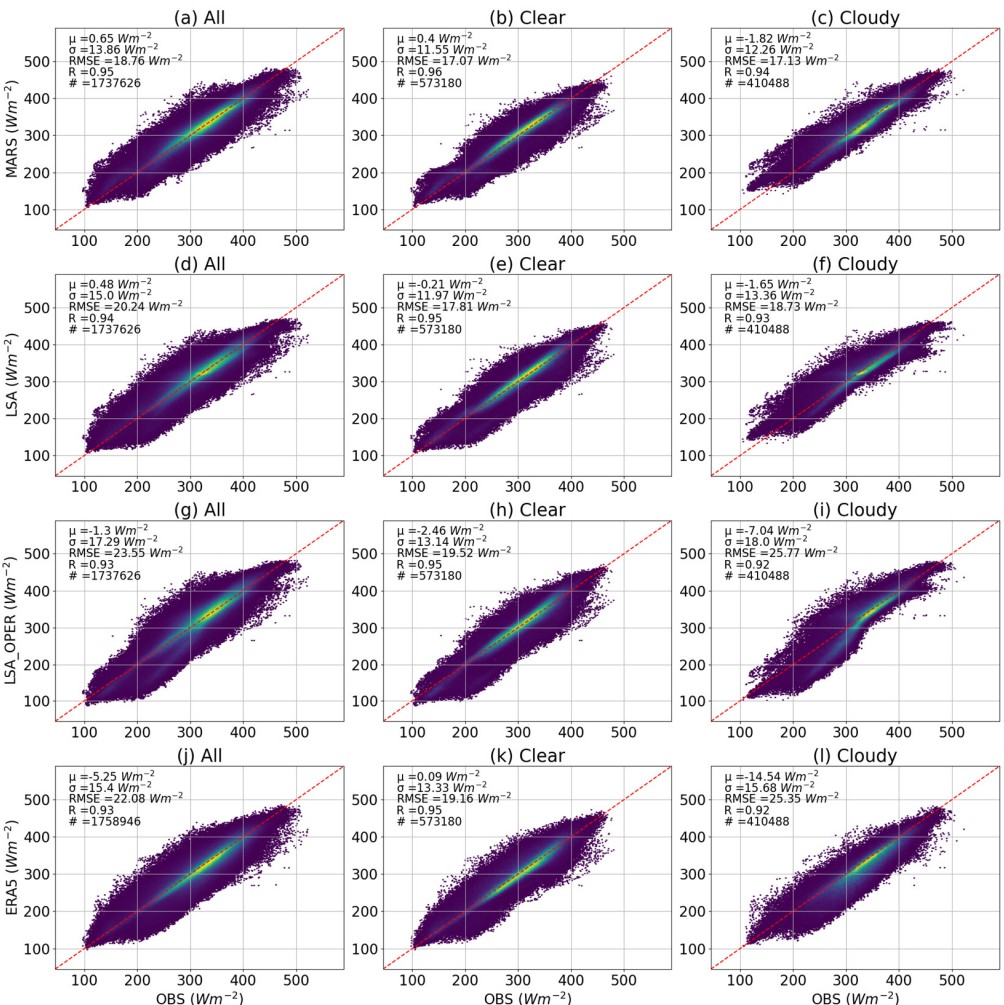

**Figure 2.** Scatter density plots of the observed DLR fluxes (W·m$^{-2}$) in the horizontal axis versus the modelled fluxes in the vertical axis for all (left), clear (middle) and cloudy (right) sky conditions estimated with 4 models: MARS (**a**–**c**), LSA (**d**–**f**), LSA_OPER (**g**–**i**), and ERA5 (**j**–**l**). The evaluation metrics are shown in the top right corner of each plot, including the number of samples used (#). The data represented includes all valid observations for the 23 ground stations in the period 2004–2019. Units are in W·m$^{-2}$.

The performance of each model can also be assessed considering different DLR ranges for all-sky conditions. The following ranges were established: (UL) upper limit for values above 400 W·m$^{-2}$, with a total of 100,521 samples; (ML) middle limit for between 200–400 W·m$^{-2}$, with a total of 1,571,217 samples; and (LL) lower limit for below 200 W·m$^{-2}$, with a total of 65,888 samples. Figure 1a gives a reasonable overview of the actual conditions represented by those ranges of DLR: there are values above 400 W·m$^{-2}$ under warm and very moist conditions, such as those found in the tropics, while the other end of the range, with DLR below 200 W·m$^{-2}$, is only found under very cold and dry conditions, such as in high latitudes winter. Table 3 includes a summary of the results for all-sky conditions considering all the data (ALL) and the median (MED) of the metrics distribution when computed for each station. Moreover, as complementary material to these results, detailed information concerning the statistics found in each station under all, clear, and cloudy-sky conditions is provided in Appendix C (Tables A5–A7, respectively). The highest performance is found in all models within the "middle" range, with similar

metrics to those computed when considering the entire dataset. This is expected due to a higher sampling, which impacts the training of the models prior to the evaluation. On the other hand, focusing on the upper and lower limits, a clear reduction of all model's performance is found. MARS and LSA underestimate the most extreme conditions, namely at higher values of DLR, while LSA_OPER and ERA5 shows a systematic underestimation of DLR in all conditions with an exception to the ERA5 overestimation at lower values. The large biases in MARS and LSA in the upper and lower conditions lead to high RMSE, with LSA_OPER showing a better overall performance. This is likely associated with the small sampling of these extreme conditions in the training dataset initially used. Despite the limitation in extreme conditions, these results are favorable to the MARS model. When pulling all stations together, the results will be dominated by those stations with larger temporal extend. This could potentially hide some problematic stations (or regions), which will be partially addressed in the following analysis.

**Table 3.** Comparison of bias ($\mu$), standard deviation of the error ($\sigma$), root mean square error (RMSE), and temporal correlation coefficient (R) between different models (MARS, LSA, LSA_OPER, and ERA5) and observations from all 23 ground stations for all-sky (2004–2019) in different conditions: considering all data (ALL); observations with values above 400 W·m$^{-2}$ (UL); observations with values between 200–400 W·m$^{-2}$ (ML); observations with values below 200 W·m$^{-2}$ (LL); and the median of the distribution of the metrics computed independently for each station (MED). Units are in W·m$^{-2}$, while correlations are given between 0–1.

| | **MARS** | | | | **LSA** | | | |
|---|---|---|---|---|---|---|---|---|
| **Condition** | $\mu$ | $\sigma$ | **RMSE** | **R** | $\mu$ | $\sigma$ | **RMSE** | **R** |
| ALL | 0.65 | 13.86 | 18.76 | 0.95 | 0.48 | 15.00 | 20.24 | 0.94 |
| UL | −9.13 | 12.02 | 18.54 | 0.61 | −11.65 | 13.74 | 21.51 | 0.54 |
| ML | 0.51 | 13.53 | 18.35 | 0.92 | 0.77 | 14.78 | 19.96 | 0.91 |
| LL | 18.82 | 14.52 | 26.97 | 0.69 | 12.21 | 15.70 | 24.47 | 0.70 |
| MED | 0.40 | 12.27 | 16.96 | 0.91 | 0.87 | 13.29 | 18.52 | 0.91 |
| | **LSA_OPER** | | | | **ERA5** | | | |
| **Condition** | $\mu$ | $\sigma$ | **RMSE** | **R** | $\mu$ | $\sigma$ | **RMSE** | **R** |
| ALL | −1.30 | 17.29 | 23.55 | 0.93 | −5.25 | 15.40 | 22.08 | 0.93 |
| UL | −2.47 | 13.37 | 17.76 | 0.57 | −11.91 | 14.34 | 22.49 | 0.52 |
| ML | −0.86 | 17.53 | 23.91 | 0.90 | −5.31 | 15.37 | 22.07 | 0.90 |
| LL | −9.87 | 14.50 | 22.56 | 0.73 | 6.44 | 15.51 | 21.73 | 0.70 |
| MED | 0.78 | 14.05 | 19.53 | 0.89 | −5.81 | 13.88 | 20.65 | 0.88 |

The performance of each model in estimating hourly DLR in each station can be assessed through the distributions of the various metrics displayed, as shown in Figure 3 boxplots for all, clear, and cloudy-sky conditions (i.e., left, middle, and right column, respectively). Each boxplot has a reference at the top, corresponding to the median value (also shown in Table 3) found for each error metric (i.e., bias, standard deviation, RMSE and correlation coefficient) and each model. Additionally, Figure 3 shows the same boxplots for the LSA* and MARS* models, which will be discussed in the next subsection for the assessment of the cloud information in DLR estimation. The results are qualitatively consistent with the previous analysis, when all data was merged, with MARS always showing better adjustments to observations (being followed by LSA, LSA_OPER, and ERA5). However, quantitatively, the median of station metrics differs from the metric considering all the data. A clear example is the temporal correlations in cloudy conditions with median values ranging between 0.86 for MARS and 0.82 for ERA5 (Figure 3l), which varied between 0.94 for MARS (Figure 2c) and 0.92 for ERA5 (Figure 2l) when considering the full data. Similarly, for the RMSE, the all-sky median varies between 16.96 in MARS and 20.65 W·m$^{-2}$ in ERA5 (Figure 3g), while it varied between 18.76 in MARS (Figure 2a) and 22.08 W·m$^{-2}$ in ERA5 (Figure 2j) when considering the full data. Moreover, the graphical

display of the metrics distribution also allows to clearly identify a better performance in clear conditions (RMSE and correlation) when compared with cloudy conditions in all models. This is associated with the different radiative impact of clouds, in particular cloud base, which is not considered in the LSA and MARS models, and limitations due to model uncertainty in ERA5. Finally, it is worth noting the presence of outliers in all estimates, which are due to several factors that can affect model accuracy in a group of stations, leading to higher deviations from observations, as shown next.

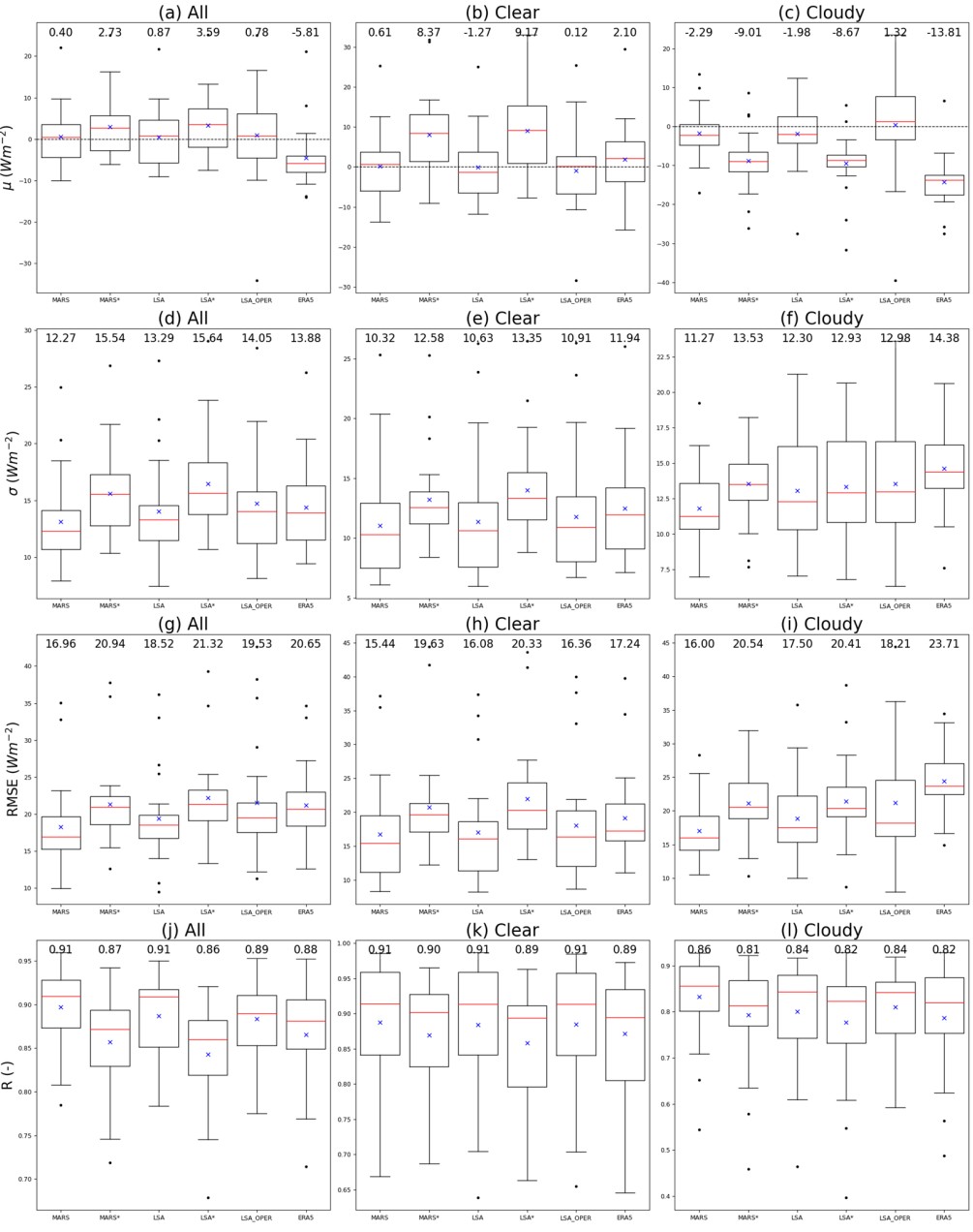

**Figure 3.** Distribution of the metrics computed for each station displayed as boxplots for all, clear, and cloudy-sky conditions (i.e., left, middle, and right column, respectively): bias, $\mu$ in (**a–c**), standard deviation of the error, $\sigma$ in (**d–f**), root mean square error, RMSE in (**g–i**), and the correlation coefficient, R in (**j–l**). The red line and blue cross inside each boxplot identify the median and mean of the distribution, respectively, with the boxes extending from the 25th to the 75th percentiles and whiskers 1.5 times the interquartile range. Units are in W·m$^{-2}$, while correlations are given between 0–1.

In addition to these results, and as a parallel validation of the MARS model, we used a set of (52) stations from an independent network with DLR observations from FLUXNET2015 [45]. The results, shown in detail in Appendix D, demonstrate the MARS model consistency between the different networks. Similarly to Table 3, in Table A9 it is possible to observe that, despite an overall error increase in all models metrics, MARS has the best performance. The best scores are found in the "middle range" for all models, where MARS presents the lowest bias and RMSE of 0.06 and 18.32 W·m$^{-2}$, respectively. The same behavior is observed when considering all data from FLUXNET2015, as well as when using data from each individual station. As previously noted, larger errors are also found at the lower and upper limits in all models. It is important to note that such results, beside reinforcing the proposed methodology, suggest BSRN observations are more appropriate for the MARS training (and validation) within the MSG-disk than the FLUXNET2015 network. BSRN operates exclusively for continuous radiation measurements at surface, where most sites provide both downward longwave and shortwave fluxes, following high standards in terms of instrument calibration and observations quality checks [41], while FLUXNET2015 targets a broader set of observations aimed at characterizing exchanges of energy, water, and carbon between the surface and the atmosphere, where available radiation plays an important role. Within FLUXNET2015, and despite the quality checks performed on measurements, availability of a complete set of measured variables (including longwave and shortwave radiation fluxes) is considered crucial [45]; across-variable quality checks are regularly performed, but the BSRN standards for radiation flux observations may not be always followed. The geographical distribution of FLUXNET2015 sites with acceptable quality radiation fluxes is limited to Europe, if we only consider sites within the MSG-disk as opposed to BSRN. These aspects are confirmed by the FLUXNET2015 validation, as depicted by the overall error increase in all models.

The evaluation procedure continues, now focusing on different case studies to highlight several aspects (positive and negative) of the different models. As previously mentioned, there are stations (most noticeably GVN, SMS, and SON) from which model estimates deviate further from observations. On the other hand, there are also stations (e.g., CAR and TAM) in which models have an overall good correspondence with observations. The following examples presented in Figure 4 depict the behavior of each model during a 36 h-period in such stations, which also include the NIM station, due to particular atmospheric effects that occur in the region. The DLR time-series of each station are shown at the hourly resolution from the different models and observations, as well as the cloud information (in the bottom subplot) from ERA5 (*tcc*) and MSG (*cf*). For the best performance cases (Figure 4a,b), the MARS model is the one that has better adjustments to observations, while ERA5 produces higher deviations. A suitable example is the CAR station (Figure 4a), where a good relation is found between the observations' DLR variability and the MSG *cf*, reflected in the MARS, LSA, and LSA_OPER simulations, while ERA5 DLR shows some deviations associated with *tcc* variability. In TAM station (Figure 4b), the reduced cloud variability clearly leads to lower deviations between models and observations, in which DLR values are found between 250–350 W·m$^{-2}$ during this time of the year. When analyzing NIM station (Figure 4c), underestimation of DLR occurs in all models. Despite a slightly higher deviation in comparison with the LSA estimates, MARS shows a smoother variation than the former, closer to the observed behavior. Regarding the worst performances cases (Figure 4d–f), significant deviations are observed in all models. In GVN station (Figure 4d), all models deviated from the observations, missing the increase in DLR at the start of the period and underestimating DLR in the following hours. Such behavior can be explained by the fact that GVN is in Antarctica, at a very high latitude, near the MSG-disk limit (close to 80°), posing significant challenges to the identification of cloudy pixels under very high view angles and under circumstances that make it difficult to separate the signature of clouds from those of snow or ice in SEVIRI/MSG observations. In SON and SMS stations, an overall overestimation of models estimates towards observations is visible, particularly in SMS (Figure 4f). For the case of SON station, the measuring equipment is located at a

relatively high altitude (about 3109 m), which, similarly to Izaña station in Tenerife, can be measuring DLR values above clouds, instead of recording values below cloud-base height. In SMS, the frequent occurrence of stratiform and shallow convective clouds [56] can lead to higher deviations, since under such conditions there is a higher difficulty to model DLR.

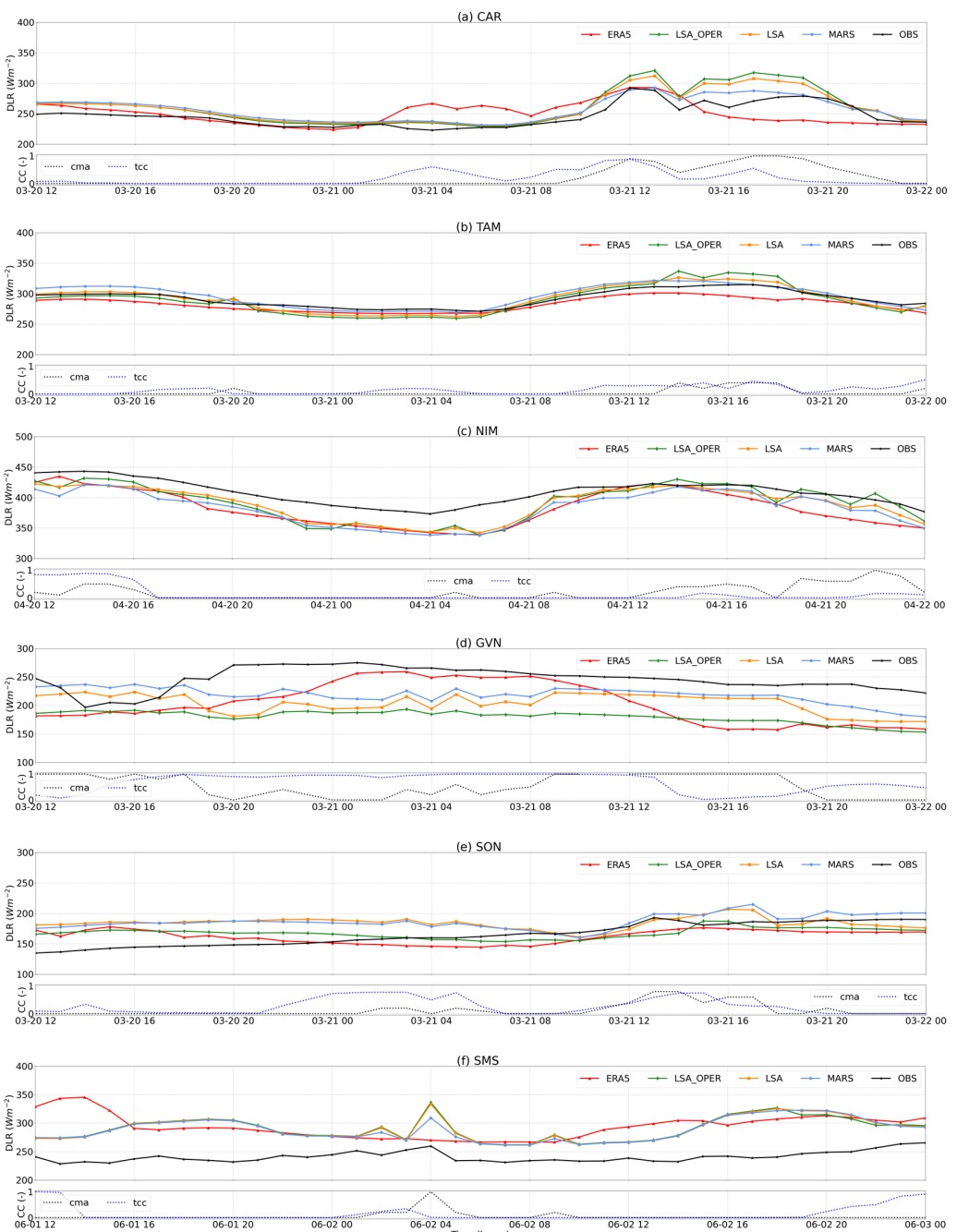

**Figure 4.** DLR (W·m$^{-2}$) hourly time-series for different ground measuring stations: CAR (**a**), TAM (**b**), NIM (**c**), GVN (**d**), SON (**e**), and SMS (**f**). The study focusses a 36 h-window between 12 and 00 UTC for each station for different periods. Different models are shown: MARS (in blue), LSA (in orange), LSA_OPER (in green), and ERA5 (in red); together with in situ observations (in black). Hourly cloud cover (CC) information is also added in the inside plots with the total cloud cover (*tcc*) from the ERA5 (blue dots) and the cloud fraction (*cf*) from the MSG (black dots).

### 3.2. Impact of Satellite Information

Following the previous results describing MARS and LSA performances, the added value that the satellite cloud information has in the calculation of DLR in both MARS and LSA algorithms is clearly observed through the MARS* and LSA* results (Figure 3). In comparison to all the other models, MARS* and LSA* stand out with an overall error increase in all the metrics. This can be primarily associated with cloud misrepresentation in ERA5. However, MARS* still shows an overall better performance than the LSA* model, although with small differences. For instance, for all-sky conditions, a correlation of 0.87 and 0.86 is found for MARS* and LSA*, respectively, while 0.88 is obtained with ERA5 (Figure 3j). These deviations are generally higher than the ERA5 estimates due to the linear relation between clear/cloudy and *tcc* that both MARS and LSA algorithms consider, which does not happen in ERA5. Following the results obtained for different ranges of DLR in the remaining models, a similar behavior of both LSA* and MARS* models is observed in Table 4. Despite an overall error increase due to the use of the *tcc* to calculate DLR, lower deviations and higher correlations (0.88 and 0.89, respectively) occur in the "middle" limit.

**Table 4.** Bias ($\mu$), standard deviation of the error ($\sigma$), root mean square error (RMSE) and temporal correlation coefficient (R) between MARS and LSA models hourly estimates using ERA5 cloud information (MARS* and LSA*) against observations from all 23 ground stations for all-sky (2004–2019). Different conditions are considered for analysis: all data (ALL); observations with values above 400 W·m$^{-2}$ (UL); observations with values between 200–400 W·m$^{-2}$ (ML); observations with values below 200 W·m$^{-2}$ (LL); and the median of the distribution of the metrics computed independently for each station (MED). Units are in W·m$^{-2}$, while correlations are given between 0–1.

| Condition | MARS* | | | | LSA* | | | |
|---|---|---|---|---|---|---|---|---|
| | $\mu$ | $\sigma$ | RMSE | R | $\mu$ | $\sigma$ | RMSE | R |
| ALL | 3.07 | 16.68 | 22.05 | 0.93 | 3.21 | 17.79 | 23.16 | 0.92 |
| UL | −9.39 | 13.87 | 20.59 | 0.53 | −11.26 | 14.55 | 22.14 | 0.51 |
| ML | 3.07 | 16.44 | 21.73 | 0.89 | 3.58 | 17.73 | 23.02 | 0.88 |
| LL | 22.01 | 15.92 | 30.25 | 0.70 | 16.48 | 16.85 | 27.61 | 0.71 |
| MED | 2.73 | 15.54 | 20.94 | 0.87 | 3.59 | 15.64 | 21.32 | 0.86 |

## 4. Discussion

The present work focusses the estimation of DLR fluxes at surface using MARS combined with hourly observations of DLR (from BSRN and ARM stations), ERA5 atmospheric profiles (*tcwv*, *t2m*, and *d2m*), and the MSG *cf*. The fact that ground and remote sensed observations are used for model training under different sky conditions provides a novel approach to estimate DLR. Similarly, to all NWP models, despite an overall good result in comparison to other models estimates, the proposed MARS model also has a few limitations. The main source of uncertainty is related to the adopted training procedure, particularly to the 23 ground stations used, which provide some degree of data availability differential. This means that stations with higher samplings will induce a local bias dependency. Moreover, the selected stations are mainly distributed in Europe, which also creates a regional dependency. When using a spatiotemporal independent set of 52 ground stations from FLUXNET2015 to validate the MARS model over a period of about 11 years, it was possible to observed that, regardless of its limitations, the proposed methodology is consistent. Most of the FLUXNET2015 stations used are located in Europe (i.e., a total of 48 stations), which limits a more global assessment of the results. Nevertheless, the MARS model continues to demonstrate an overall better estimation of DLR when compared with the remaining models. Furthermore, we should keep in mind that in situ observations may also be subject to significant uncertainties. Other sources of error can result from the *cf* information used for the sky classification, in which the data quality is dependent on the satellite interpretation of clouds and associated errors.

Considering all the available data, the validation of the MARS model (Figure 2) shows that, generally, and despite its limitations, using measured data for training (i.e., a total of about 5.86 years) purposes produces better adjustments to observed values. This is the case for the MARS (Figure 2a–c) and LSA (Figure 2d–f) model estimates. Particularly, MARS provides the best performance under the different sky conditions as a result of using an automatic piecewise regression method instead of the least square fitting method used in the LSA-SAF original algorithm. In comparison, the worst results are found with ERA5, which also depict an overall decrease in performance. As previously mentioned, the ERA5 negative bias in cloudy conditions might be partially explained by problems in the representation of clouds and their radiative effect. It is worth noting that this effect is observed not only during cloudy-sky periods (where higher deviations are found) but also during clear-sky periods (with a very low bias of $-0.46$ W·m$^{-2}$), in which ERA5 is likely assuming some situations of cloud occurrence when none are to be observed, thus leading to overestimation that lowers bias close to zero. Moreover, the separation between clear and cloudy conditions is performed using the satellite information, which can introduce a few inconsistencies regarding the actual observations (due to satellite footprint and uncertainties in cloud detection) and ERA5 atmospheric conditions (e.g., cloud base errors). Therefore, the interpretation of the model's evaluation in clear versus cloudy conditions is not straightforward. Nevertheless, in the absence of accurate cloud information from NWP models, satellite information should be used instead of reanalysis data for model training purposes, as particularly shown by the improved DLR estimates of MARS in comparison to the ones found with MARS*. Another overall underestimation of DLR is similarly found with the LSA-SAF operational model (LSA_OPER), although with smaller deviations than in the ERA5 model. In that case, the poorer performance must be attributed to the original calibration carried out by Trigo et al. [13], where TIGR-like and MODTRAN-4 simulations (not observations) were used to calibrate the model parameters (Table A1). Additionally, a common feature of the original LSA-SAF algorithm is the 'S' shape curve in DLR estimates, as shown by the LSA_OPER results (Figure 2g–i), where its effects are particularly visible under cloudy conditions. Since the plots of MARS, the newly calibrated LSA model, and ERA5 do not present that characteristic, it is likely an artifact introduced by the MODTRAN-4 estimates used in the fitting. It should be noted that, although MARS eliminates most of the previous error signatures from LSA-SAF and ERA5 models, there is still room for improvement, particularly for cloudy conditions. In particular, the way to further incorporate satellite observations related to, e.g., cloud type and cloud phase, is still largely unexplored.

The results analyzed so far suggest an overall good performance of the considered models, but also reveal the presence of several outliers in all model estimates (Figure 3). Taking into account the information provided by Tables A5–A7, it is possible to find significant deviations towards observations in three stations (GVN, SMS, and SON), either due to the latitude or altitude effect, as well as measurement inaccuracies related to equipment malfunction. Figure 4 presents (36-h) examples of the behavior in each model for a selected group of very different stations, which are aimed to represent the best- and worst-case studies. Despite the good results in CAR and TAM stations (Figure 4a,b), it is important to consider the fact that the spatial sampling is not equally distributed throughout all the selected stations within the MSG-disk. In terms of the spatial distribution, the use of observations for validation allows to test the performance of different model estimates over different climate regions, strengthening the validation of the proposed formulation. However, as previous mentioned, regional dependencies should be expected in regions that have a higher number of stations (e.g., Europe), therefore contributing to an overall bias reduction to the MARS and LSA models. Nevertheless, for the best-case studies, ERA5 continues to produce higher deviations due deficiencies in cloud representation. When analyzing NIM station (Figure 4c), a clear underestimation of measured values is provided by all models (between 340–450 W·m$^{-2}$). This aspect can be explained with the fact that NIM station may be subject to high aerosol loads, usually desert dust, which can lead to

a significant increase in observed DLR when compared to similar aerosol free conditions, which is not captured well by any model. In particular, this station usually experiences higher occurrence of extreme dust events (e.g., desert storms), resulting in larger deviations of estimations from observations [44]. For the worst-case studies, the latitude effect in GVN (Figure 4d), the altitude effect in SON (Figure 4e), and the measuring inaccuracies in SMS (Figure 4f) result in very high errors between estimated and observed values, particularly in the former. At a very high latitude, the LSA_OPER seems to provide a higher deviation from observations, which is in accordance with one of the LSA-SAF model limitation stated by Trigo et al. [14], while the MARS and LSA models demonstrate to have a better approximation to observations, particularly MARS.

## 5. Conclusions

This work aimed at contributing to a new and improved formulation for the estimation of downward long-wave radiation (DLR) at the surface. The new formulation was built considering the combination of hourly reanalysis, ground, and remote observed inputs to train a state-of-the-art machine learning algorithm based on multivariate adaptive regression splines (MARS). The use of satellite data not only allows us to perform better estimates of DLR with suitable temporal and spatial samplings under different sky conditions, but also provides a wide spatial coverage at high-resolution.

When compared with the Satellite Application Facility on Land Surface Analysis (LSA-SAF) algorithm, results showed that the MARS algorithm performs very well, providing better adjustments to observed DLR fluxes than the former, where lower errors and higher correlations were evident under all, clear, and cloudy-sky conditions. This is mainly related to the fact that MARS allows to replace the previous least square fitting criterion for a set of pre-defined atmospheric states implemented in the LSA-SAF with a more refined discretization that accounts with the best fitting option based on maximum reduction on sum-of-squares residual error. Systematic differences and an overall underestimation were found in both LSA_OPER and ERA5 models, being linked to the original calibration with the Thermodynamic Initial Guess Retrieval (TIGR, [38]) atmospheric-profile database and the Moderate Resolution Atmospheric Transmittance and Radiance Code (MODTRAN-4, [37]) fluxes, and the cloud representation by the total cloud cover retrieved from ERA5, respectively. The role of satellite information in the calculation of DLR was also evaluated using both MARS and LSA models but considering ERA5 cloud information instead of the satellite cloud information to separate clear and cloudy situations (MARS* and LSA* models). The results clearly showed the added value of using remotely sensed data instead of reanalysis cloud cover.

The evaluation analysis, performed within the MSG-disk (i.e., longitude/latitude within 75°E/N), continued to show that MARS provided best results in comparison to the remaining models (LSA, LSA_OPER, and ERA5). In particular, the use of ground observations, from the baseline surface radiation network (BSRN, [41]) and the atmospheric radiation measurement (ARM, [42]) user facility, to calibrate MARS led to improved adjustments with lower errors and higher correlations (in which sampling plays an important role). During the validation procedure it was shown that, when using all available data from the 23 stations, MARS allows us to obtain RMSEs of 18.76, 17.07, and 17.13 W·m$^{-2}$ under all, clear, and cloudy conditions, respectively. Lower errors were found when considering the performance of the model at each measuring location, as shown by the median values of the RMSE for all, clear, and cloudy-sky, i.e., 16.96, 15.44, and 16.00 W·m$^{-2}$, respectively. Moreover, the reduction and elimination of previous systematic differences and overall underestimation carried out by LSA_OPER and ERA5 was achieved. The added value of using the satellite cloud information was accessed by comparing with estimates driven by ERA5 total cloud cover, showing an increase of 17% of the RMSE. Finally, the proposed methodology was further validated against independent observations gathered from 52 FLUXNET2015 ground stations over an 11–year period, showing that MARS DLR estimates have a better approximation to observations than the remaining models.

There is potential in using the proposed MARS formulation for operational purposes, however there are still a few steps towards improvement that need to be carried out in the future. These include: (i) the assessment of DLR estimates on a regional level, by producing regional maps and comparing MARS estimates with LSA-SAF product outputs (a fundamental procedure in order to operationalize MARS estimates); (ii) improvements of MARS estimates with enhanced input fields from ECMWF numerical weather prediction (e.g., increased resolution, better model physics, data assimilation); and (iii) other MARS model variants that can make use of other satellite products, such as measurements of thermal infrared bands and the top of atmosphere radiances as inputs for the training phase, similarly to Zhou et al. [23].

An application example for the estimation of hourly DLR values from the MARS model is made available in the Supplementary Materials, including a python code and the two calibrated MARS submodels (i.e., for clear and cloudy skies), and a synthetic test data for a 24-h period.

**Supplementary Materials:** The following supporting information can be downloaded at: https://www.mdpi.com/article/10.3390/rs14071704/s1; Code S1: zip compressed archive with 3 files: (i) MARS clear-sky model: mars_bsrn_model_clear_sky.sav; (ii) MARS cloudy-sky model: bsrn_model_cloud_sky.sav and (iii) example python script to run the MARS model and calculate DLR at surface using synthetic input data: MARS_DLR_output_2020.py.

**Author Contributions:** Conceptualization, E.D. and I.F.T.; Methodology, E.D., I.F.T. and F.M.L.; Software, F.M.L.; Validation, F.M.L.; Formal Analysis, F.M.L.; Investigation, E.D., I.F.T. and F.M.L.; Resources, E.D. and I.F.T.; Data Curation, F.M.L.; Writing—Original Draft Preparation, F.M.L.; Writing—Review & Editing, F.M.L., E.D. and I.F.T.; Visualization, F.M.L.; Supervision, E.D. and I.F.T.; Project Administration, E.D.; Funding Acquisition, E.D. All authors have read and agreed to the published version of the manuscript.

**Funding:** This work was performed within the framework of the LSA-SAF (https://landsaf.ipma.pt/en/; accessed on 31 March 2022) project, funded by EUMETSAT, and by the European Union Horizon 2020 research and innovation program No 958927.

**Data Availability Statement:** Public and private datasets were analyzed in this study. However, data supporting the results are available from the corresponding author upon reasonable request.

**Acknowledgments:** The authors are thankful for the availability of Copernicus and ECMWF in providing the needed data extracted from ERA5 through the climate data store website (https://cds.climate.copernicus.eu; accessed on 31 March 2022), and access to the BSRN and ARM station data. F.M.L. acknowledges the funding by Fundação para a Ciência e a Tecnologia (FCT) grant number PTDC/CTA-MET/28946/2017 (CONTROL), and from European Union Horizon 2020 research and innovation program under grant agreement No 958927 (CoCO2).

**Conflicts of Interest:** The authors declare no conflict of interest. The sponsors had no role in the design, execution, interpretation, or writing of the study.

**Appendix A. LSA-SAF Algorithm**

In the LSA-SAF algorithm, DLR ($F^{\downarrow}$) is estimated through a bulk parameterization given by the following equations, as described by Trigo et al. [14]:

$$F^{\downarrow} = \sigma \epsilon_{sky} T_{sky}^4, \tag{A1}$$

where $\sigma$ is the Stefan–Boltzmann constant, $\epsilon_{sky}$ and $T_{sky}^4$ are the sky effective emissivity and the sky effective temperature, respectively. The former is given as a function of the total column of water vapor (*tcwv*), as follows:

$$\epsilon_{sky} = 1 - \left[ 1 + \left( \frac{tcwv}{10} \right) \exp\left( -\left( \alpha + \beta \frac{tcwv}{10} \right)^m \right) \right], \tag{A2}$$

where *m* are values that best adjust to clear and cloudy conditions (0.5 and 1, respectively). For the case of the latter, the following relation is used:

$$T_{sky} = T_0 + (\delta\Delta d_0 + \gamma), \tag{A3}$$

where $T_0$ is the 2-metre temperature corrected through the 2-metre observed dewpoint depression ($\delta\Delta d_0$). The parameters $\alpha$, $\beta$, $\gamma$, and $\delta$ in Equations (A2) and (A3) are fitted independently for cloudy-sky and clear-sky conditions, and the all-sky DLR is the sum of the clear $F_{clear}^{\downarrow}$) and cloudy $F_{cloudy}^{\downarrow}$) contributions considering the cloud fraction (*cf*):

$$DLR = cf\, F_{cloudy}^{\downarrow} + (1 - cf)F_{clear}^{\downarrow}. \tag{A4}$$

More information regarding the calibration procedure of the parameters in Equations (A2) and (A3) is presented by Trigo et al. [13]. The method follows a piecewise regression independent for clear and cloudy conditions considering three classes of profiles: (i) dry cold, with *tcwv* ≤ 10 mm and *t2m* < 270 K; (ii) dry and warm, with *tcwv* ≤ 10 mm and *t2m* > 270 K; and (iii) moist, with *tcwv* > 8 mm. For the calibration phase of the operational LSA-SAF algorithm (LSA_OPER) atmospheric profiles from the TIGR-like database, namely the *tcwv*, *t2m*, *d2m*, and fluxes simulated with MODTRAN-4, were used. Moreover, the separation of clear and cloudy skies in the calibration database considers the total cloud cover (*tcc*), where clear and cloudy conditions are assigned for *tcc* = 0 and *tcc* > 0.9, respectively, in which a piecewise regression method is then applied to each set of clear and cloudy conditions, separately. The parameters used by the current LSA-SAF operational algorithm (LSA_OPER) are presented in Table A1.

**Table A1.** Calibrated parameters for the LSA_OPER model [14], i.e., the LSA-SAF operational algorithm that makes use of TIGR-like database (1992–1993) [38] for different atmospheric profiles under clear and cloudy-sky.

| | **Clear-Sky** | | | | **Cloudy-Sky** | | | |
|---|---|---|---|---|---|---|---|---|
| **Profiles** | $\alpha$ | $\beta$ | $\gamma$ | $\delta$ | $\alpha$ | $\beta$ | $\gamma$ | $\delta$ |
| Dry Cold | 0.653 | 4.796 | 1.253 | −0.739 | 0.968 | 2.257 | −0.236 | −0.877 |
| Dry Warm | 0.704 | 3.720 | 1.655 | −0.151 | 3.446 | 0.369 | 0.278 | −0.443 |
| Moist | 0.587 | 3.344 | 1.686 | −0.203 | 3.446 | 0.369 | 0.278 | −0.443 |

**Table A2.** Calibrated parameters for the LSA model using ERA5 inputs and observed DLR (BSRN and ARM) for the different atmospheric profiles under clear and cloudy-sky.

| | **Clear-Sky** | | | | **Cloudy-Sky** | | | |
|---|---|---|---|---|---|---|---|---|
| **Profiles** | $\alpha$ | $\beta$ | $\gamma$ | $\delta$ | $\alpha$ | $\beta$ | $\gamma$ | $\delta$ |
| Dry Cold | 2.289 | 4.992 | −2.368 | −1.129 | 1.804 | 3.026 | 0.436 | −0.991 |
| Dry Warm | 0.865 | 3.701 | 0.532 | −0.135 | 3.229 | 0.324 | 0.737 | −0.562 |
| Moist | 1.466 | 3.051 | 0.5709 | −0.187 | 3.229 | 0.324 | 0.737 | −0.562 |

## Appendix B. Models Training

For the training of both LSA and MARS models (i.e., for clear and cloudy conditions), the full dataset was divided in two components: training and verification. The training dataset was constructed by randomly selecting 40% of the full-time series for each station limited to a maximum of 6 months of data used from each station, with the remaining data being used as the verification dataset. This corresponded to a total of 51,386 (5.87 years) and 932,282 (106.42 years) hourly samples for the training and verification period, respectively. The bias and root mean square error (RMSE) for clear and cloudy conditions in the training and verification samples are presented in Tables A3 and A4, respectively. The results show that MARS always performs better than LSA under clear and cloudy-sky conditions.

During the training stage, MARS presents a very small bias (close to zero) as opposed to LSA, which shows a relatively large bias in comparison with higher deviation for the training under clear-sky. The RMSE, besides being lower in MARS (although with less discrepancies than the bias found for each model), does not vary significantly from clear to cloud sky. A similar behavior is observed for the verification phase, despite an overall increase in the bias in both models, which is related to the sampling increase with a higher data availability differential.

**Table A3.** Bias and root mean square error of LSA-SAF and MARS for the training and verification samples (#) in clear-sky conditions. Units are in $W \cdot m^{-2}$.

|  | Training (#31820) | | Verification (#541360) | |
| :---: | :---: | :---: | :---: | :---: |
| **Models** | **Bias** | **RMSE** | **Bias** | **RMSE** |
| LSA | 0.24 | 20.41 | −0.27 | 18.99 |
| MARS | −0.00 | 19.71 | 0.12 | 18.00 |

**Table A4.** Bias and root mean square error of LSA-SAF and MARS for the training and verification samples (#) in cloudy-sky conditions. Units are in $W \cdot m^{-2}$.

|  | Training (#19566) | | Verification (#390922) | |
| :---: | :---: | :---: | :---: | :---: |
| **Models** | **Bias** | **RMSE** | **Bias** | **RMSE** |
| LSA | 1.20 | 21.71 | −1.06 | 19.95 |
| MARS | −0.00 | 19.36 | −1.55 | 18.19 |

**Appendix C. Evaluation Detailed Results**

The following tables comprise all the statistical error metrics obtained. Tables A5–A7 show the scores for each station in all, clear, and cloudy-sky conditions (respectively).

**Table A5.** Error metrics between model (MARS, LSA_OPER, LSA, and ERA5) and measuring station (all-sky conditions) for 2004–2019. Bias (μ), standard deviation (σ), and root mean square error (RMSE) are in W·m$^{-2}$; temporal correlation coefficient. (R) is given between 0–1.

| | μ | | | | σ | | | | RMSE | | | | R | | | |
|---|---|---|---|---|---|---|---|---|---|---|---|---|---|---|---|---|
| **Station** | **MARS** | **LSA** | **LSA_OPER** | **ERA5** | **MARS** | **LSA** | **LSA_OPER** | **ERA5** | **MARS** | **LSA** | **LSA_OPER** | **ERA5** | **MARS** | **LSA** | **LSA_OPER** | **ERA5** |
| BRB | 5.72 | 5.77 | 10.33 | 1.46 | 9.36 | 10.10 | 10.84 | 11.22 | 13.26 | 14.03 | 17.09 | 14.99 | 0.92 | 0.91 | 0.91 | 0.88 |
| BUD | 0.40 | 1.14 | 6.66 | −1.80 | 7.93 | 7.44 | 8.13 | 9.42 | 9.98 | 9.49 | 12.22 | 12.61 | 0.91 | 0.92 | 0.91 | 0.86 |
| CAB | −0.41 | −0.39 | 0.00 | −7.09 | 11.84 | 13.29 | 13.65 | 13.96 | 15.60 | 17.33 | 17.99 | 20.65 | 0.93 | 0.91 | 0.91 | 0.89 |
| CAM | 0.44 | 0.87 | 2.79 | −4.58 | 15.38 | 16.61 | 15.88 | 16.96 | 19.87 | 21.39 | 21.03 | 23.33 | 0.87 | 0.85 | 0.85 | 0.82 |
| CAR | 3.60 | 4.09 | 5.86 | −8.04 | 9.74 | 10.60 | 11.09 | 11.38 | 13.57 | 14.89 | 15.98 | 17.89 | 0.96 | 0.95 | 0.95 | 0.94 |
| CNR | 0.34 | 3.96 | 4.75 | −0.11 | 12.84 | 13.43 | 14.22 | 15.83 | 16.88 | 18.52 | 20.01 | 21.07 | 0.91 | 0.90 | 0.89 | 0.86 |
| DAA | 5.71 | 5.45 | 6.38 | −4.24 | 11.62 | 12.69 | 12.77 | 10.46 | 16.27 | 17.44 | 18.07 | 16.28 | 0.94 | 0.93 | 0.93 | 0.93 |
| ENA | −8.78 | −7.89 | −3.89 | −6.24 | 14.35 | 14.34 | 13.96 | 16.78 | 20.12 | 20.15 | 19.07 | 22.69 | 0.84 | 0.84 | 0.83 | 0.77 |
| FLO | −6.96 | −7.09 | −1.79 | −7.28 | 10.80 | 10.95 | 11.00 | 12.43 | 15.73 | 15.92 | 14.73 | 18.14 | 0.91 | 0.91 | 0.91 | 0.88 |
| GAN | 3.52 | 5.90 | 14.38 | −9.07 | 15.18 | 18.55 | 18.99 | 13.87 | 21.77 | 25.44 | 29.06 | 22.53 | 0.90 | 0.87 | 0.86 | 0.91 |
| GOB | −7.61 | −6.99 | −6.68 | −8.86 | 11.35 | 12.17 | 12.72 | 11.62 | 18.72 | 19.07 | 19.53 | 19.54 | 0.88 | 0.88 | 0.88 | 0.88 |
| GVN | −1.70 | −7.07 | −34.06 | −10.88 | 18.49 | 20.28 | 21.99 | 18.75 | 23.23 | 26.68 | 42.63 | 27.26 | 0.88 | 0.85 | 0.85 | 0.86 |
| NIM | −9.96 | −9.00 | −4.59 | −13.84 | 11.88 | 12.01 | 14.58 | 14.63 | 18.48 | 18.04 | 18.72 | 23.71 | 0.93 | 0.94 | 0.94 | 0.92 |
| LIN | 3.59 | 2.38 | 0.78 | −3.88 | 12.76 | 14.03 | 15.68 | 13.88 | 16.96 | 18.14 | 20.34 | 20.05 | 0.93 | 0.91 | 0.90 | 0.90 |
| PAL | 0.69 | 0.14 | 0.77 | −4.62 | 12.27 | 13.43 | 14.05 | 14.20 | 16.19 | 17.49 | 18.60 | 20.87 | 0.92 | 0.91 | 0.90 | 0.88 |
| PAR | −1.20 | −3.02 | 4.90 | −4.58 | 9.09 | 8.34 | 8.28 | 10.38 | 11.29 | 10.72 | 11.30 | 13.45 | 0.78 | 0.82 | 0.83 | 0.71 |
| PAY | 0.60 | 5.22 | 2.72 | −6.25 | 13.02 | 13.18 | 15.58 | 17.01 | 17.24 | 18.67 | 22.06 | 23.97 | 0.92 | 0.91 | 0.89 | 0.85 |
| PTR | 7.89 | 9.75 | 16.61 | 8.06 | 9.79 | 10.17 | 10.15 | 12.87 | 14.87 | 16.21 | 21.04 | 18.69 | 0.86 | 0.85 | 0.86 | 0.77 |
| SBO | −6.85 | −4.39 | −4.37 | −5.81 | 13.82 | 14.13 | 14.90 | 13.71 | 19.44 | 19.66 | 20.70 | 19.52 | 0.88 | 0.87 | 0.85 | 0.88 |
| SMS | 22.07 | 21.70 | 25.10 | 21.12 | 20.34 | 22.17 | 21.93 | 20.40 | 35.04 | 36.19 | 38.23 | 34.65 | 0.85 | 0.83 | 0.83 | 0.85 |
| SON | 9.73 | 1.81 | −9.88 | −5.08 | 24.98 | 27.29 | 28.44 | 26.26 | 32.76 | 33.03 | 35.67 | 33.05 | 0.81 | 0.78 | 0.78 | 0.79 |
| TAM | −6.15 | −8.79 | −6.82 | −14.05 | 10.55 | 12.64 | 11.33 | 10.41 | 14.95 | 18.62 | 16.14 | 20.36 | 0.96 | 0.94 | 0.95 | 0.95 |
| TOR | −2.67 | −2.30 | −8.38 | −7.80 | 13.91 | 14.73 | 18.50 | 14.38 | 18.39 | 19.35 | 25.14 | 21.86 | 0.93 | 0.92 | 0.90 | 0.91 |

**Table A6.** Error metrics between model (MARS, LSA_OPER, LSA, and ERA5) and measuring station (clear-sky conditions) for 2004–2019. Bias (μ), standard deviation (σ), and root mean square error (RMSE) are in W·m$^{-2}$; temporal correlation coefficient. (R) is given between 0–1.

| | μ | | | | σ | | | | RMSE | | | | R | | | |
|---|---|---|---|---|---|---|---|---|---|---|---|---|---|---|---|---|
| **Station** | **MARS** | **LSA** | **LSA_OPER** | **ERA5** | **MARS** | **LSA** | **LSA_OPER** | **ERA5** | **MARS** | **LSA** | **LSA_OPER** | **ERA5** | **MARS** | **LSA** | **LSA_OPER** | **ERA5** |
| BRB | 4.29 | 5.41 | 5.03 | 5.36 | 6.81 | 7.45 | 8.49 | 8.05 | 9.84 | 11.00 | 11.91 | 12.22 | 0.91 | 0.91 | 0.90 | 0.89 |
| BUD | −4.58 | −2.89 | 0.89 | −1.71 | 6.12 | 6.46 | 6.89 | 7.79 | 8.92 | 8.48 | 8.71 | 11.06 | 0.94 | 0.94 | 0.94 | 0.89 |
| CAB | −1.45 | −1.53 | −2.19 | 2.10 | 6.47 | 6.54 | 6.85 | 12.02 | 9.52 | 9.77 | 10.15 | 16.40 | 0.97 | 0.97 | 0.97 | 0.92 |

**Table A6.** *Cont.*

| | μ | | | | σ | | | | RMSE | | | | R | | | |
|---|---|---|---|---|---|---|---|---|---|---|---|---|---|---|---|---|
| Station | MARS | LSA | LSA_OPER | ERA5 | MARS | LSA | LSA_OPER | ERA5 | MARS | LSA | LSA_OPER | ERA5 | MARS | LSA | LSA_OPER | ERA5 |
| CAM | 3.44 | 2.32 | 1.88 | 8.64 | 12.99 | 13.11 | 13.20 | 14.97 | 19.02 | 19.03 | 18.99 | 22.86 | 0.84 | 0.83 | 0.83 | 0.80 |
| CAR | 2.27 | 2.68 | 2.23 | −3.58 | 6.19 | 5.99 | 6.73 | 7.99 | 8.31 | 8.24 | 8.96 | 11.61 | 0.99 | 0.99 | 0.98 | 0.97 |
| CNR | 1.58 | 1.96 | 1.27 | 10.87 | 8.55 | 8.62 | 8.97 | 11.63 | 12.49 | 12.71 | 12.98 | 18.88 | 0.96 | 0.96 | 0.96 | 0.93 |
| DAA | 6.10 | 5.01 | 3.14 | −2.00 | 10.80 | 11.11 | 11.18 | 7.69 | 15.44 | 15.34 | 14.97 | 12.55 | 0.93 | 0.93 | 0.93 | 0.94 |
| ENA | −9.59 | −10.46 | −8.86 | 2.51 | 15.21 | 15.40 | 15.02 | 15.77 | 21.49 | 22.05 | 21.12 | 21.00 | 0.83 | 0.83 | 0.83 | 0.79 |
| FLO | −7.48 | −7.56 | −5.23 | −0.83 | 10.82 | 10.80 | 11.22 | 12.89 | 16.74 | 16.82 | 16.36 | 16.93 | 0.91 | 0.91 | 0.91 | 0.89 |
| GAN | −9.00 | −6.89 | −5.11 | −7.03 | 13.76 | 14.28 | 15.30 | 14.54 | 21.73 | 21.45 | 21.87 | 21.50 | 0.86 | 0.86 | 0.86 | 0.86 |
| GOB | −7.55 | −6.56 | −7.70 | −7.22 | 10.32 | 11.16 | 11.90 | 10.15 | 17.87 | 18.20 | 19.14 | 17.24 | 0.88 | 0.87 | 0.87 | 0.89 |
| GVN | −2.98 | −11.82 | −28.34 | −6.23 | 20.37 | 23.89 | 23.62 | 17.12 | 25.50 | 30.77 | 40.02 | 25.13 | 0.73 | 0.64 | 0.65 | 0.75 |
| NIM | −13.80 | −10.86 | −10.66 | −15.68 | 10.83 | 10.99 | 13.70 | 12.90 | 19.94 | 18.11 | 20.50 | 23.45 | 0.94 | 0.95 | 0.95 | 0.93 |
| LIN | 4.05 | 4.10 | 2.93 | 6.48 | 6.86 | 7.01 | 6.95 | 10.75 | 10.68 | 10.92 | 10.46 | 16.54 | 0.98 | 0.98 | 0.98 | 0.95 |
| PAL | 0.61 | 0.36 | −0.22 | 3.02 | 7.72 | 7.91 | 8.05 | 11.94 | 11.50 | 11.74 | 11.94 | 17.16 | 0.96 | 0.96 | 0.96 | 0.92 |
| PAR | 1.53 | −1.27 | 6.95 | 4.78 | 8.12 | 6.94 | 7.22 | 8.07 | 10.92 | 9.86 | 12.16 | 11.60 | 0.67 | 0.70 | 0.70 | 0.65 |
| PAY | 2.96 | 3.34 | 2.17 | 6.17 | 7.98 | 7.87 | 8.05 | 13.92 | 13.10 | 13.11 | 13.03 | 19.67 | 0.96 | 0.96 | 0.96 | 0.91 |
| PTR | 8.74 | 12.74 | 16.27 | 12.14 | 7.34 | 7.70 | 8.48 | 11.04 | 13.41 | 16.68 | 19.92 | 18.84 | 0.85 | 0.85 | 0.85 | 0.78 |
| SBO | −7.54 | −6.36 | −7.90 | −3.65 | 12.86 | 12.83 | 13.26 | 11.92 | 18.61 | 18.30 | 19.20 | 17.19 | 0.90 | 0.90 | 0.89 | 0.91 |
| SMS | 25.26 | 24.99 | 25.40 | 29.40 | 19.07 | 19.65 | 19.67 | 19.20 | 37.13 | 37.40 | 37.64 | 39.75 | 0.79 | 0.78 | 0.79 | 0.81 |
| SON | 12.64 | 8.16 | 0.12 | 9.71 | 25.32 | 26.27 | 26.30 | 26.00 | 35.46 | 34.25 | 33.05 | 34.50 | 0.71 | 0.71 | 0.72 | 0.74 |
| TAM | −2.67 | −4.41 | −7.61 | −11.25 | 8.53 | 8.93 | 9.33 | 7.14 | 11.43 | 12.17 | 14.15 | 15.20 | 0.96 | 0.96 | 0.96 | 0.97 |
| TOR | −2.22 | −2.36 | −5.75 | 1.69 | 10.64 | 10.63 | 10.91 | 13.48 | 16.22 | 16.08 | 17.27 | 19.24 | 0.95 | 0.95 | 0.96 | 0.94 |

**Table A7.** Error metrics between model (MARS, LSA_OPER, LSA, and ERA5) and measuring station (cloudy-sky conditions) for 2004–2019. Bias (μ), standard deviation (σ), and root mean square error (RMSE) are in W·m$^{-2}$; temporal correlation coefficient. (R) is given between 0–1.

| | μ | | | | σ | | | | RMSE | | | | R | | | |
|---|---|---|---|---|---|---|---|---|---|---|---|---|---|---|---|---|
| Station | MARS | LSA | LSA_OPER | ERA5 | MARS | LSA | LSA_OPER | ERA5 | MARS | LSA | LSA_OPER | ERA5 | MARS | LSA | LSA_OPER | ERA5 |
| BRB | 3.30 | 1.59 | 9.20 | −9.94 | 8.37 | 9.04 | 9.33 | 11.73 | 11.44 | 12.08 | 15.33 | 17.96 | 0.71 | 0.64 | 0.62 | 0.62 |
| BUD | 5.35 | 4.01 | 11.75 | −6.80 | 7.96 | 7.30 | 7.03 | 10.53 | 11.33 | 10.00 | 14.66 | 14.86 | 0.83 | 0.83 | 0.84 | 0.63 |
| CAB | −2.74 | −2.99 | −3.17 | −15.53 | 10.19 | 11.57 | 12.34 | 13.88 | 14.12 | 15.97 | 17.61 | 23.71 | 0.90 | 0.88 | 0.87 | 0.87 |
| CAM | −0.65 | 0.19 | 2.94 | −13.56 | 12.36 | 13.67 | 12.98 | 15.42 | 17.20 | 18.85 | 18.17 | 24.27 | 0.84 | 0.83 | 0.83 | 0.83 |
| CAR | 1.09 | 0.67 | 4.45 | −17.54 | 11.11 | 12.78 | 12.65 | 14.36 | 15.38 | 17.50 | 18.16 | 25.45 | 0.90 | 0.87 | 0.86 | 0.87 |
| CNR | −2.92 | 3.69 | 4.49 | −12.52 | 11.33 | 12.5 | 13.70 | 15.38 | 15.80 | 17.54 | 20.34 | 23.32 | 0.86 | 0.83 | 0.81 | 0.83 |
| DAA | −0.10 | 0.08 | 8.82 | −17.90 | 13.54 | 17.71 | 16.71 | 15.34 | 18.20 | 23.23 | 23.57 | 26.65 | 0.90 | 0.85 | 0.86 | 0.89 |
| ENA | −10.56 | −7.51 | −0.98 | −17.58 | 10.98 | 9.55 | 9.61 | 14.11 | 17.84 | 15.21 | 13.36 | 25.28 | 0.81 | 0.84 | 0.84 | 0.75 |

**Table A7.** *Cont.*

| | μ | | | | σ | | | | RMSE | | | | R | | | |
|---|---|---|---|---|---|---|---|---|---|---|---|---|---|---|---|---|
| Station | MARS | LSA | LSA_OPER | ERA5 | MARS | LSA | LSA_OPER | ERA5 | MARS | LSA | LSA_OPER | ERA5 | MARS | LSA | LSA_OPER | ERA5 |
| FLO | −7.35 | −8.22 | −1.46 | −13.58 | 8.25 | 7.53 | 7.49 | 10.73 | 12.82 | 12.78 | 9.86 | 19.42 | 0.86 | 0.88 | 0.88 | 0.81 |
| GAN | 6.77 | 12.47 | 23.50 | −12.51 | 14.42 | 20.12 | 19.22 | 13.12 | 21.07 | 28.25 | 33.82 | 22.67 | 0.93 | 0.92 | 0.92 | 0.93 |
| GOB | −9.19 | −11.47 | −1.60 | −25.76 | 13.65 | 12.30 | 12.05 | 16.55 | 19.73 | 21.17 | 17.07 | 33.16 | 0.84 | 0.83 | 0.84 | 0.82 |
| GVN | −2.29 | −4.06 | −39.48 | −17.53 | 14.04 | 14.65 | 16.32 | 20.61 | 18.76 | 19.76 | 44.43 | 31.10 | 0.86 | 0.85 | 0.85 | 0.77 |
| NIM | −0.76 | −4.19 | 6.61 | −11.05 | 11.78 | 14.27 | 12.31 | 16.87 | 16.00 | 19.23 | 17.45 | 24.68 | 0.79 | 0.67 | 0.73 | 0.70 |
| LIN | −0.27 | −1.98 | −6.62 | −12.94 | 11.15 | 11.91 | 14.79 | 13.53 | 15.18 | 16.54 | 21.41 | 22.27 | 0.90 | 0.89 | 0.87 | 0.88 |
| PAL | −2.19 | −3.03 | −3.53 | −13.81 | 10.50 | 11.09 | 12.60 | 14.38 | 14.26 | 15.52 | 18.21 | 23.39 | 0.90 | 0.88 | 0.86 | 0.86 |
| PAR | −5.96 | −5.65 | 1.32 | −13.61 | 6.99 | 7.05 | 6.34 | 7.60 | 10.48 | 10.38 | 7.93 | 16.66 | 0.54 | 0.46 | 0.59 | 0.49 |
| PAY | −4.00 | 3.42 | −2.22 | −16.68 | 11.07 | 11.75 | 16.22 | 16.95 | 16.13 | 16.88 | 23.52 | 27.53 | 0.89 | 0.88 | 0.85 | 0.83 |
| PTR | −3.92 | −3.20 | 6.48 | −7.99 | 8.84 | 8.19 | 7.77 | 12.73 | 12.04 | 11.14 | 12.06 | 18.07 | 0.65 | 0.61 | 0.66 | 0.56 |
| SBO | −4.71 | 4.08 | 11.83 | −19.26 | 15.42 | 18.68 | 18.15 | 16.46 | 20.48 | 24.02 | 25.58 | 28.74 | 0.83 | 0.75 | 0.77 | 0.80 |
| SMS | 13.41 | 10.98 | 16.51 | 6.54 | 16.26 | 17.80 | 18.12 | 16.11 | 25.61 | 26.29 | 29.20 | 23.01 | 0.78 | 0.73 | 0.73 | 0.81 |
| SON | 9.83 | −0.99 | −16.69 | −15.14 | 19.23 | 21.27 | 23.62 | 20.31 | 28.30 | 29.40 | 36.30 | 31.49 | 0.77 | 0.70 | 0.73 | 0.76 |
| TAM | −17.03 | −27.47 | −9.71 | −27.56 | 13.16 | 17.99 | 15.31 | 15.86 | 23.98 | 35.80 | 22.34 | 34.43 | 0.93 | 0.88 | 0.91 | 0.89 |
| TOR | −4.93 | −4.37 | −13.86 | −14.48 | 11.27 | 11.7 | 16.83 | 13.37 | 15.89 | 16.67 | 26.48 | 23.21 | 0.92 | 0.90 | 0.89 | 0.89 |

## Appendix D. FLUXNET2015 Validation

The following results show the use of FLUXNET2015 [45] dataset for the validation of the MARS model. To this end, 30-min data from 52 ground stations within the MSG-disk were aggregated to hourly values and then used for validation purposes considering a period of 11 years between 2004 and 2015 (Table A8). In Table A9, a summary of the error metrics distribution for different ranges of DLR and under all-sky conditions is shown.

**Table A8.** List of 52 stations from FLUXNET2015 used within the Meteosat Second Generation (MSG) disk for validation purposes of estimated downward long-wave radiation (DLR) at the surface. The name, acronym, location, geographical coordinates (°), elevation (m), availability (total number of years available between 2004 and 2015), and annual mean DLR (W·m$^{-2}$) for each station is shown.

| Station | Acronym | Location | Latitude and Longitude (°) | Elev. (m) | Avail. (Years) | Annual DLR (W.m$^{-2}$) |
|---|---|---|---|---|---|---|
| Neustift | AT-Neu | Austria | 47.12°N; 11.32°E | 970 | 6.96 | 288.86 |
| Brasschaat | BE-Bra | Belgium | 51.31°N; 4.52°E | 16 | 7.33 | 322.16 |
| Lonzee | BE-Lon | Belgium | 50.55°N; 4.75°E | 167 | 7.38 | 320.71 |
| Chamau | CH-Cha | Switzerland | 47.21°N; 8.41°E | 393 | 9.29 | 321.45 |
| Davos | CH-Dav | Switzerland | 46.82°N; 9.86°E | 1639 | 7.70 | 273.69 |
| Früebüel | CH-Fru | Switzerland | 47.12°N; 8.54°E | 982 | 8.80 | 306.25 |
| Laegern | CH-Lae | Switzerland | 47.48°N; 8.36°E | 689 | 9.25 | 304.40 |
| Oensingen grassland | CH-Oe1 | Switzerland | 47.29°N; 7.73°E | 450 | 4.87 | 326.28 |
| Oensingen crop | CH-Oe2 | Switzerland | 47.29°N; 7.73°E | 452 | 10.66 | 326.24 |
| Bily Kriz forest | CZ-BK1 | Czech Republic | 49.50°N; 18.54°E | 875 | 7.23 | 313.66 |
| Bily Kriz grassland | CZ-BK2 | Czech Republic | 49.49°N; 18.54°E | 855 | 5.32 | 313.12 |
| Trebon | CZ-wet | Czech Republic | 49.03°N; 14.77°E | 426 | 7.82 | 322.84 |
| Anklam | DE-Akm | Germany | 53.87°N; 13.68°E | −1 | 4.52 | 320.87 |
| Gebesee | DE-Geb | Germany | 51.10°N; 10.92°E | 162 | 11.00 | 307.30 |
| Grillenburg | DE-Gri | Germany | 50.95°N; 13.51°E | 385 | 8.07 | 311.01 |
| Hainich | DE-Hai | Germany | 51.08°N; 10.45°E | 430 | 8.84 | 310.49 |
| Klingenberg | DE-Kli | Germany | 50.89°N; 13.52°E | 478 | 10.58 | 303.74 |
| Lackenberg | DE-Lkb | Germany | 49.10°N; 13.31°E | 1308 | 3.60 | 296.89 |
| Leinefelde | DE-Lnf | Germany | 51.33°N; 10.37°E | 451 | 5.96 | 306.42 |
| Oberbärenburg | DE-Obe | Germany | 50.79°N; 13.72°E | 734 | 6.90 | 301.53 |
| Rollesbroich | DE-RuR | Germany | 50.62°N; 6.30°E | 515 | 3.48 | 318.72 |
| Selhausen Juelich | DE-RuS | Germany | 50.87°N; 6.45°E | 103 | 2.65 | 330.62 |
| Schechenfilz Nord | DE-SfN | Germany | 47.81°N; 11.33°E | 590 | 2.44 | 323.40 |
| Spreewald | DE-Spw | Germany | 51.89°N; 14.03°E | 61 | 4.33 | 322.56 |
| Tharandt | DE-Tha | Germany | 50.96°N; 13.57°E | 385 | 10.74 | 311.30 |
| Zarnekow | DE-Zrk | Germany | 53.88°N; 12.89°E | 0 | 1.61 | 328.96 |
| Soroe | DK-Sor | Denmark | 55.49°N; 11.65°E | 40 | 6.81 | 314.17 |
| Hyytiala | FI-Hyy | Finland | 61.85°N; 24.30°E | 181 | 4.25 | 306.04 |
| Lompolojankka | FI-Lom | Finland | 67.99°N; 24.21°E | 274 | 2.93 | 282.32 |
| Grignon | FR-Gri | France | 48.84°N; 1.95°E | 125 | 10.80 | 328.80 |
| Le Bray | FR-LBr | France | 44.72°N; 0.77°W | 61 | 5.01 | 333.98 |
| Puechabon | FR-Pue | France | 43.74°N; 3.60°E | 270 | 9.11 | 318.07 |
| Guyaflux | GF-Guy | French Guiana | 5.28°N; 52.93°W | 48 | 2.00 | 411.47 |
| Ankasa | GH-Ank | Gana | 5.27°N; 2.69°W | 124 | 2.00 | 405.94 |
| Borgo Cioffi | IT-BCi | Italy | 40.52°N; 14.96°E | 20 | 4.12 | 331.83 |
| Castel d'Asso1 | IT-CA1 | Italy | 42.38°N; 12.03°E | 200 | 3.35 | 341.22 |
| Castel d'Asso2 | IT-CA2 | Italy | 42.38°N; 12.03°E | 200 | 2.59 | 345.66 |
| Castel d'Asso3 | IT-CA3 | Italy | 42.38°N; 12.02°E | 197 | 2.90 | 339.64 |
| Collelongo | IT-Col | Italy | 41.85°N; 13.59°E | 1560 | 7.35 | 280.26 |
| Ispra ABC-IS | IT-Isp | Italy | 45.81°N; 8.63°E | 210 | 2.00 | 335.75 |
| Lavarone | IT-Lav | Italy | 45.96°N; 11.28°E | 1353 | 10.43 | 289.81 |
| Monte Bondone | IT-MBo | Italy | 46.02°N; 11.05°E | 1550 | 8.97 | 282.15 |
| Arca di Noe | IT-Noe | Italy | 40.61°N; 8.15°E | 25 | 9.50 | 349.74 |
| Renon | IT-Ren | Italy | 46.59°N; 11.43°E | 1730 | 8.84 | 280.63 |
| Roccarespampani 1 | IT-Ro1 | Italy | 42.49°N; 11.93°E | 235 | 1.00 | 310.57 |

**Table A8.** *Cont.*

| Station | Acronym | Location | Latitude and Longitude (°) | Elev. (m) | Avail. (Years) | Annual DLR (W.m$^{-2}$) |
|---|---|---|---|---|---|---|
| Roccarespampani 2 | IT-Ro2 | Italy | 42.39°N; 11.92°E | 160 | 1.51 | 332.02 |
| Torgnon | IT-Tor | Italy | 45.84°N; 7.58°E | 2160 | 5.81 | 274.69 |
| Horstermeer | NL-Hor | Netherlands | 52.24°N; 5.07°E | 2 | 7.00 | 326.18 |
| Loobos | NL-Loo | Netherlands | 52.17°N; 5.74°E | 25 | 10.95 | 343.79 |
| Fyodorovskoye | RU-Fyo | Russia | 56.46°N; 32.92°E | 265 | 2.76 | 293.74 |
| Stordalen grassland | SE-St1 | Sweden | 68.35°N; 19.05°E | 351 | 1.99 | 297.90 |
| Mongu | ZM-Mon | Zambia | 15.44°S; 23.25°E | 1053 | 1.85 | 358.19 |

**Table A9.** Comparison of bias (μ), standard deviation of the error (σ), root mean square error (RMSE), and temporal correlation coefficient (R) between different models (MARS, LSA, LSA_OPER, and ERA5) and observations from all 52 FLUXNET2015 ground stations for all-sky (2004–2015) in different conditions: considering all data (ALL); observations with values above 400 W·m$^{-2}$ (UL); observations with values between 200–400 W·m$^{-2}$ (ML); observations with values below 200 W·m$^{-2}$ (LL); and the median of the distribution of the metrics computed independently for each station (MED). Units are in W·m$^{-2}$, while correlations are given between 0–1.

| | MARS | | | | LSA | | | |
|---|---|---|---|---|---|---|---|---|
| Condition | μ | σ | RMSE | R | μ | σ | RMSE | R |
| ALL | 0.05 | 18.49 | 24.14 | 0.88 | 0.86 | 19.01 | 24.66 | 0.87 |
| UL | −16.91 | 15.53 | 26.45 | 0.41 | −16.79 | 14.99 | 25.82 | 0.42 |
| ML | 0.06 | 18.32 | 23.90 | 0.85 | 0.77 | 18.76 | 24.35 | 0.84 |
| LL | 20.51 | 16.75 | 30.81 | 0.39 | 26.68 | 16.07 | 34.78 | 0.41 |
| MED | 0.42 | 16.40 | 22.52 | 0.87 | 1.16 | 16.85 | 22.77 | 0.86 |
| | LSA_OPER | | | | ERA5 | | | |
| Condition | μ | σ | RMSE | R | μ | σ | RMSE | R |
| ALL | −1.60 | 20.90 | 27.24 | 0.86 | −8.71 | 18.93 | 26.71 | 0.87 |
| UL | −9.94 | 15.94 | 23.07 | 0.41 | −20.90 | 17.74 | 30.77 | 0.37 |
| ML | −1.63 | 21.05 | 27.40 | 0.82 | −8.92 | 18.76 | 26.55 | 0.84 |
| LL | 9.95 | 16.37 | 24.82 | 0.46 | 15.32 | 18.24 | 28.79 | 0.38 |
| MED | −2.02 | 18.83 | 25.88 | 0.85 | −8.08 | 16.29 | 23.82 | 0.86 |

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
