# Peer review of "Integrating Reanalysis and Satellite Cloud Information to Estimate Surface Downward Long-Wave Radiation"

_remotesensing, doi:10.3390/rs14071704_

Round 1
Reviewer 1 Report
The manuscript proposes a new downward longwave radiation estimation method using a MARS model by integrating site, reanalysis, and remote sensing data. After the revision, the manuscript well addressed my concerns, and I would recommend it to be published after minor editing.
1) Line 9: The 'paramount' looks too strong for readers. Suggest replacing it with 'very important' or some other words else.
Author Response
We have corrected the reviewer's comment.

Reviewer 2 Report
Most of the problems raised have been solved.
Author Response
The reviewer had no final suggestions.

This manuscript is a resubmission of an earlier submission. The following is a list of the peer review reports and author responses from that submission.
Round 1
Reviewer 1 Report
The purpose of this paper is to propose a new and improved algorithm to estimate long-wave radiation (DLR) underground. Innovative innovations in combining hourly reanalysis, coupled ground-based, remote observation data, and machine learning algorithms have broken through the limitations of sky conditions, making satellite data more spatially applicable in the DLR estimated direction. This research has some significance, but there are also some problems.
- The title of the article is too long. I suggest simplifying the title.
- For the handling of missing and abnormal data, which is only re-summarized according to the hourly frequency, can you consider another method for handling missing and abnormal values? Considering the completeness of the data may be better. (Lines 169-177)
- Part of the DLR estimated by Mars algorithm is driven by ERA5 cloud information. Is it appropriate to only compare the estimation results with the ground station data that has not participated in the test in space? Or it can be compared with the data in the future outside the test to make the model more convincing.
- The description of the algorithm model and the overall experimental process could be deepened regarding, such as the flowchart of the image, the algorithm steps, etc., This will make your research more convincing!
- Distribution of the metrics computed for each station displayed as boxplots for all, clear and cloudy sky conditions,What is the significance of the display divided into different sites? Or is there any indication of the advantages of the algorithm?
- The description of the new model is not sufficient, there is no drawing; There are many machine learning algorithms in SK-learn. Can it be compared with other machine learning algorithms? In addition, this paper focuses on algorithms, so it is too simple to compare with LSA-SAF. Whether model innovation needs to be further improved.
- Page 13: Which can be related to equipment malfunction or absence of sensor calibration. Is it possible to interpret deviations as "possibly related to something or other"? The mechanism and mechanism of the deviation can be further strengthened.
Author Response
All comments made by the reviewer are answered in detail in the attached file.

Reviewer 2 Report
This study proposes a MARS-based method for estimating downward longwave radiation from ground measurements, reanalysis, and satellite cloud information. The methodology is reasonable, and assessment results are comprehensive and promising to me, however, some major issues are necessary to be dealt with before the publication. My concerns are about the spatial representativeness of the training samples and the assessment strategy. Please find them below.
Major
1) Literature review about machine learning (ML)-based DLR estimation and highlight your innovation
Even the authors have summarized the basic methodologies for estimating DLR in the Introduction, the ML-based literatures are not focused. Considering that there are so many ML-based studies have been published in recent years, I would recommend using one paragraph to comprehensively summarize the pros and cons of the previous ML-based studies for estimating DLR, not just limited to MARS-based paper in Discussion. A comprehensive summary will be another contribution of this work.
Also, I would recommend moving the literature review part in Discussion to Introduction, and then you can highlight your innovations and what differences can you make by comparing with previous ML studies. Just changing an ML model, like a random forest to MARS, won’t be a considerable innovation.
2) My major concern is about the limited spatial sampling (23 sites) for modeling compared to the wide study area (Europe, Africa, and South America), even this issue has been mentioned in the Discussion.
This study utilized a large number of samples for training because a long timespan is included, not due to a wide spatially sampling coverage. Climate is stable at one place, and the relationship won’t change significantly between DLR and input variables in different years. Therefore, simply increasing the sampling period won’t bring more information to the model and cannot improve the representativeness and robustness of the model, especially when the authors claim to implement the model for data production. I would recommend including more sites from FLUXNET2015 or other European Flux networks to increase the spatial sampling.
3) assessment strategy: the samples for calibration/modeling and verification should be independent, spatially, and temporally.
The strategy of this study combined all available samples and divided them randomly, which is not rational enough, especially when the site spatial coverage is not enough compared to the large study area. If the validation is not spatiotemporally independent, the validation statistics should be very good as you’ve included so many years in these locations.
I would recommend that in each cross-validation, 20-30% sites are randomly selected in 2-3 years (e.g., 2017-2019) for validation, independently; and the rest 70% sites in 2004-2016 are for training. Samples of the validation sites in 2004-2016 (or samples of the training sites in 2017-2019) cannot be used for modeling (validation). More sites are necessary, or you may include another network only as a validation dataset.
Minor
Line 7: please check the requirement of the journal if it should be written as ‘[7]’ or ‘Jon Snow et al. [7]’
Line 101: uniquly -> uniquely
Line 105: [13], similar issue as [7]
Line 122: does ‘calibrate’ here mean ‘train’? ‘calibrate’ is often used for correcting some biases that its unit won’t change. Building/Training a model is usually to estimate a dependent from many other variables.
Line 132: ‘:’ can be removed
Line 237: please check if ‘the’ is necessary before ‘Appendix A’ in the requirement
Line 236: The difference between LSA and LSA-OPER is only the different version of input data? Why the LSA_OPER doesn’t update the input from ERA40 to ERA5? And the calibration period seems very old.
Line 327: usually a ‘,’ is needed behind ‘C2’
Line 720, 732: journal name is partly missing
Author Response
All comments made by the reviwer are answered in detail in the attached file.
